# COFT: Counterfactual–Conformal Decoding for Fair Chain-of-Thought Reasoning in Large Language Models

## Abstract

Large language models (LLMs) can reveal and amplify societal biases during chain-of-thought (CoT) generation. We present **COFT** (*Chain of Fair Thought*), a training-free decoding method that provides instance-level fairness control with statistical guarantees for any frozen causal language model. COFT operates in three stages. First, it creates a masked counterfactual prompt by replacing sensitive spans with neutral tokens. Second, it compares the factual and masked logit distributions through lightweight logit fusion to attenuate attribute-driven biases. Third, it uses dual-branch split-conformal calibration to certify per-step candidate token sets at a user-chosen risk level. We evaluate COFT across six models and multiple bias benchmarks. Our method reduces standard bias metrics by 30–55% (median 38%) while preserving task utility and language quality. Reasoning accuracies remain unchanged within run-to-run noise margins. The computational overhead is modest, equivalent to one additional forward pass ($\leq 11\%$). COFT offers a clear, auditable path to safer CoT generation with significant bias reduction, negligible utility loss, and no requirement for retraining, auxiliary classifiers, or weight access.

## 1 Introduction

Large–scale autoregressive language models (LMs) now underpin open-ended generation, question answering, and conversational assistants (Brown et al., 2020; Kojima et al., 2022; Touvron et al., 2023). They are trained by next-token prediction on large web corpora and, as a result, exhibit emergent abilities such as in-context learning and chain-of-thought (CoT) reasoning (Dong et al., 2022; Wei et al., 2024). The same corpora, however, encode historical power imbalances and social stereotypes, which LMs can internalize and even amplify (Bender et al., 2021; Zhao et al., 2021). When models generate explicit CoT, these biases need not remain latent: harmful associations can appear token by token in the reasoning trace, even when the final answer looks neutral.

Existing mitigation strategies address parts of this problem but leave important gaps. Data curation and domain-specific fine-tuning can suppress some harmful behaviors, but they require expensive retraining and may degrade general-domain performance or encode the biases of annotators (Gehman et al., 2020; Santurkar et al., 2023). Inference-time steering provides an alternative that avoids changing model weights. Methods based on auxiliary classifiers or expert ensembling can push generation away from unsafe content (Madotto et al., 2020; Liu et al., 2021), but they inherit the blind spots of the classifiers they rely on and introduce extra computational latency. A third line of work performs global representation-space debiasing by removing linear attribute subspaces from model representations (Ravfogel et al., 2020; 2022). Because this nullspace is fixed, it cannot adapt to the semantics of a specific prompt, and it can suppress legitimate content or fail to capture non-linear manifestations of bias (Liang et al., 2023).

Two critical desiderata therefore remain insufficiently addressed. *First*, most approaches lack instance-level statistical guarantees at decode time. For a given prompt, they do not specify which tokens can be safely emitted while keeping the risk of failure under control. Conformal prediction (CP) provides a way to obtain such guarantees under minimal assumptions (Vovk et al., 2005; Angelopoulos & Bates, 2021). CP works by calibrating prediction sets to a user-chosen risk level. Adapting CP to autoregressive decoding, however, is non-trivial: one must define nonconformity scores that respect sequence dependence and the open-vocabulary nature of language generation (Zhao et al., 2023). *Second*, fairness goals are often defined only at a global or aggregate level. In real deployments, we instead need a local notion of counterfactual parity. The model's output should be stable under

hypothetical changes to sensitive spans in the prompt, and this stability should hold at each decoding step (Chiappa, 2019).

What is currently missing is a decoding-time mechanism that satisfies three properties at once: (i) it enforces counterfactual invariance at the level of candidate next tokens, (ii) it is gradient-free and model-agnostic so that it can be used with frozen checkpoints, and (iii) it provides auditable, per-instance guarantees that are suitable for interactive systems.

We introduce **COFT**—*Chain of Fair Thought*—as a post–training, decode-time intervention that performs *counterfactual inference at test time*. Instead of relying on "fairness through unawareness," COFT explicitly constructs a counterfactual version of each prompt. It does so by masking sensitive spans with a neutral sentinel token (e.g., [MASK]). Both the original (factual) prompt and the masked (counterfactual) prompt are then passed through the *same* frozen causal LM. COFT intervenes directly in logit space. At each step of decoding, it takes the two next-token logit vectors (one from the factual view and one from the counterfactual view) and combines them to produce a compromise distribution. This mixed distribution steers predictions toward fairer outputs, without any retraining and without introducing auxiliary classifiers. The control is *autoregressive*: the intervention is applied at every decoding step, so each emitted token is generated under an explicit counterfactual fairness constraint.

To turn this attenuation into formal guarantees, COFT uses a *dual-branch split-conformal* calibration procedure on a disjoint calibration set. This yields per-step, instance-level coverage in the form of certified token sets and calibrated thresholds. We further analyze robustness under mild distribution shift using density-ratio bounds between calibration and deployment distributions. The resulting procedure is gradient-free, model-agnostic, and auditable. It adds only modest and predictable overhead: one extra masked forward pass per decoding step. In our experiments, COFT preserves benign contextual information while reducing stereotype and toxicity metrics across recent open-weight models. The full methodology and theoretical guarantees are presented in §3, and comprehensive empirical evaluations appear in §4.

## 2 BACKGROUND AND RELATED WORK

### 2.1 BIAS IN LARGE LANGUAGE MODELS

Large-scale autoregressive LMs trained on web data inherit societal stereotypes and historical power imbalances alongside genuine linguistic regularities (Bender et al., 2021). Empirical studies show biased associations across protected attributes such as gender, race, religion, and disability in both unconditional generation and downstream tasks. These biases manifest as toxicity, stereotypes, and demographic disparities (Gehman et al., 2020; Sheng et al., 2019; Blodgett et al., 2020). Benchmarks such as CrowS-Pairs, StereoSet, BBQ, and BOLD make these harms measurable using contrastive prompts, targeted QA, and open-ended continuations (Nangia et al., 2020; Nadeem et al., 2020; Parrish et al., 2021; Dhamala et al., 2021). Instruction tuning and RLHF can reduce overt toxicity, but more subtle forms of stereotyping and disparate treatment often remain. These issues are especially visible when chain-of-thought (CoT) rationales expose intermediate reasoning steps that encode stereotypes (Zhao et al., 2021). The central challenge is to mitigate such biases without incurring retraining costs or sacrificing utility and fluency, and to do so despite the sensitivity of LMs to prompt wording, decoding choices, and the open-ended nature of the output space. (Gehman et al., 2020; Santurkar et al., 2023).

### 2.2 COUNTERFACTUAL FAIRNESS

Let $A$ denote a protected attribute (e.g., gender), $\hat{Y}$ a model output, and $U$ latent factors that capture all non–protected causes of the output in a structural causal model (Pearl, 2009). A predictor is *counterfactually fair* if the following holds: for the same $U$, intervening to set $A = a'$ does not change the distribution of $\hat{Y}$ (Kusner et al., 2017; Chiappa, 2019). Path-specific variants of this notion further restrict which causal mechanisms from $A$ to $\hat{Y}$ are considered impermissible (Chiappa, 2019; Kilbertus et al., 2017).

In our decoding setting, $\hat{Y}$ is the *next token* at step $t$, and the "individual" is the current prefix $w_{<t}$, which captures the benign context seen so far. We therefore operationalize fairness as *local counterfactual parity*. Concretely, the next-token distribution should remain stable when spans conveying $A$ in the prompt are replaced with a neutral sentinel token. That is, we compare the factual prompt $p$ to its masked counterpart $\widetilde{p} = M(p)$ while holding $w_{<t}$ fixed.

This instance-level, stepwise notion of fairness (i) matches interactive generation, where outputs are consumed token by token; (ii) targets the actionable mechanism at inference time, namely the conditional next-token distribution; and (iii) enables transparent auditing by directly contrasting the factual and masked branches within the same model. This stands in contrast to purely group-level criteria such as demographic parity, which operate on aggregate outcomes rather than on individual prompts and decoding steps.

### 2.3 CONFORMAL PREDICTION: ESSENTIALS FOR GENERATIVE MODELING

Conformal prediction (CP) provides distribution-free, finite-sample guarantees by calibrating a *nonconformity score* $s(x, y)$ on a held-out *calibration* set and then selecting a ceiling-corrected $(1-\alpha)$ quantile of these scores. This procedure controls the expected miscoverage at level $\alpha$ (Vovk et al., 2005; Angelopoulos & Bates, 2021). In split (inductive) CP, the predictive model is fixed in advance. We compute scores $\{s_i\}$ on i.i.d. calibration examples, and the resulting threshold $\tau$ defines prediction sets $\{y : s(x, y) \leq \tau\}$ with marginal coverage at least $1-\alpha$ under exchangeability.

For autoregressive LMs, we use *stepwise* scores that depend on next-token probabilities or logits, so that we obtain guarantees at each decoding step. We also enforce *policy consistency*: the decoding settings used during calibration are exactly the same as those used at test time. This alignment helps preserve exchangeability of the stepwise contexts (Zhao et al., 2023). Under dataset shift, covariate-shift–aware variants of CP relate test-time miscoverage to a density-ratio inflation of $\alpha$. This link supports robustness diagnostics and simple corrections (Tibshirani et al., 2019; Barber et al., 2021; Angelopoulos & Bates, 2021). These components (score design, split calibration, and shift-aware analysis) are exactly what we leverage to control, at the instance and token level, which outputs may be safely emitted during decoding.

### 2.4 RELATED WORK

Mitigating harmful bias in LMs spans data-level, training-time, and inference-time approaches. Data filtering and augmentation methods aim to reduce harmful correlations at the source (Gehman et al., 2020; Sheng et al., 2019; Blodgett et al., 2020). Counterfactual data augmentation (CDA) swaps or masks sensitive markers to balance evidence across demographic groups (Zhao et al., 2018; Lu et al., 2020). Fine-tuning and reinforcement learning from human feedback (RLHF) adapt models toward safety or alignment objectives (Gehman et al., 2020; Santurkar et al., 2023). These routes can be effective, but they require access to model weights and substantial compute, must be repeated as data or use cases evolve, and risk degrading general-domain competence or encoding annotator preferences.

Inference-time control methods avoid retraining. Plug-and-play methods steer generation using auxiliary discriminators (Madotto et al., 2020). Expert reweighting approaches such as DExperts and GeDi shift token probabilities by combining pro-experts and anti-experts, or by using generative discriminators (Liu et al., 2021; Krause et al., 2020). Prompt-only self-debiasing methods rely on carefully designed safety prompts or templates to reduce bias (Schick et al., 2021). While practical, these methods have important limitations. Many depend on external classifiers, which add latency and import the classifiers' blind spots. They also do not provide distribution-free, instance-level guarantees. Prompt-only strategies tend to be brittle, with performance that varies significantly across models and tasks.

Representation-space debiasing methods operate on hidden states inside the model. Approaches such as INLP and adversarial training remove attribute subspaces or reduce the recoverability of protected attributes in intermediate representations (Ravfogel et al., 2020; 2022; Elazar & Goldberg, 2018). These global projections are, however, prompt-agnostic. They may inadvertently erase legitimate, context-dependent semantics (Liang et al., 2023), and they may fail to capture non-linear manifestations of bias. Most importantly, they typically require updating model weights, which limits their applicability to frozen checkpoints.

Conformal prediction (CP) provides distribution-free error control by calibrating nonconformity scores and selecting quantile-based thresholds (Vovk et al., 2005; Angelopoulos & Bates, 2021). Recent work adapts CP to large language models in several ways: stepwise or sequence-level scores for autoregressive decoding (Zhao et al., 2023; Fayyazi et al., 2025), risk-controlled rebalancing of competing objectives (e.g., toxicity vs. utility) (He et al., 2024), and validity guarantees for alignment or refusal behavior in RLHF-style systems (Wang et al., 2024; Zhang et al., 2024). Covariate-shift–aware variants relate miscoverage to density ratios between calibration and deployment distributions (Tib-

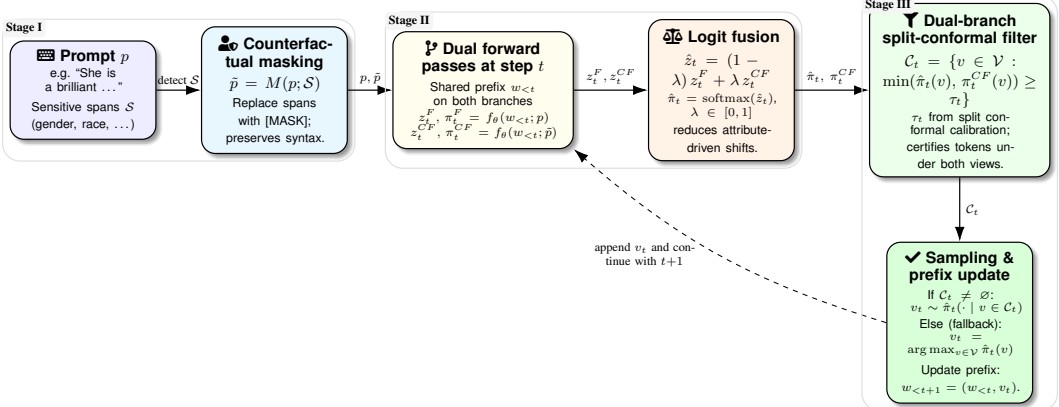

Figure 1: **COFT decoding workflow at step** $t$**.** Given a prompt $p$ and sensitive spans $\mathcal{S}$, COFT forms a masked counterfactual prompt $\tilde{p} = M(p; \mathcal{S})$, runs parallel factual/masked forward passes sharing the same prefix $w_{<t}$, fuses logits with weight $\lambda$, and applies a dual-branch split-conformal filter to obtain a certified candidate set $\mathcal{C}_t$ for the next token $v_t$.

shirani et al., 2019; Barber et al., 2021). However, existing methods are typically post hoc (e.g., re-ranking generations or calibrating refusal decisions) and do not impose explicit counterfactual fairness constraints at decode time. They therefore leave open how to regulate *token-level* decisions during decoding when sensitive information is present in the prompt, and how to combine CP with counterfactual masking and logit fusion in a single, training-free procedure.

**Our contribution.** COFT intervenes exactly where bias manifests: in the next-token distribution during decoding. For each prompt, it constructs a masked counterfactual version within the same frozen model (§3.2). It then applies a lightweight *logit fusion* step (§3.3) that attenuates attribute-driven disparities by combining factual and counterfactual logits. On top of this, COFT uses a *dual-branch split-CP* procedure (§3.4) to certify a shared support set of tokens with distribution-free, per-step guarantees. All of this is done without retraining and without auxiliary classifiers. In this way, COFT directly operationalizes local counterfactual parity during decoding while preserving the model's utility.

## 3 METHODOLOGY

We consider a frozen causal LM $f_\theta$ with tokenizer vocabulary $\mathcal{V} = \{1, \ldots, V\}$. For a prompt $p$ and a prefix $w_{<t}$, the model produces logits $z_t \in \mathbb{R}^V$ and a next-token distribution $\pi_t = \text{softmax}(z_t)$.

Let $\mathcal{S}$ be a set of *sensitive* spans (for example, gendered, racial, or religious identifiers). These spans can be specified by the user or detected automatically using *Named Entity Recognition* (NER) (Lample et al., 2016). We define a deterministic masking operator $M$ that replaces each $s \in \mathcal{S}$ with a tokenizer-stable sentinel token $[\texttt{MASK}]$, while preserving the original word order (details in §3.2).

COFT decodes two parallel views that share the same prefix: the factual prompt $p$, and the masked prompt $\tilde{p} = M(p)$. At each step, it performs two actions. First, it applies *counterfactual logit fusion*, which is a convex interpolation of the factual and masked logits controlled by a parameter $\lambda \in [0, 1]$. This fusion attenuates attribute-driven disparities (§3.3). Second, it imposes a *dual-branch split-conformal* acceptance rule, which is calibrated offline at miscoverage level $\alpha$. This rule only admits tokens that are simultaneously probable under both the factual and masked views (§3.4).

Sampling is then performed from the surgically corrected factual distribution, restricted to this set of certified candidate tokens. The procedure requires no training, gradients, or auxiliary classifiers. Figure 1 summarizes the overall workflow.

### 3.1 NOTATION AND DESIDERATA

**Two views per step.** At step $t$, COFT evaluates $f_\theta$ twice, conditioning both runs on the *same generated prefix* $w_{<t}$:

$$\underbrace{z_t^F, \pi_t^F = f_\theta(w_{<t}; p)}_{\text{factual}}, \qquad \underbrace{z_t^{CF}, \pi_t^{CF} = f_\theta(w_{<t}; \tilde{p})}_{\text{masked}}, \tag{1}$$

where $\widetilde{p} = M(p)$ replaces each span in $\mathcal{S}$ by a neutral sentinel token (defined below). Importantly, the counterfactual branch is a *probe* used to measure and attenuate sensitivity. Note that we never sample from it.

**Counterfactual fairness target (token-level).** Let $v_t^\star$ denote the ground-truth next token at step $t$ under the factual world. We say a decoder is *token-level counterfactually stable at level* $\alpha$ if, for every step $t$, the set of eligible next tokens $\mathcal{C}_t$ produced by the decoder satisfies

$$\mathbb{P}\big[\, v_t^\star \in \mathcal{C}_t \text{ under } p \,\wedge\, v_t^\star \in \mathcal{C}_t \text{ under } \widetilde{p} \,\big] \geq 1 - \alpha, \tag{2}$$

and sampling is performed *only* from $\mathcal{C}_t$ using the corrected factual distribution. This ensures that the realized token is supported by *both* worlds with miscoverage at most $\alpha$.

## 3.2 STAGE I: COUNTERFACTUAL MASKING

**Masking operator.** Given a prompt $p$ and a span set (possibly multi-token) $\mathcal{S}$, define the masking operator $M :$ text $\rightarrow$ text that deterministically replaces each span $s \in \mathcal{S}$ with a neutral sentinel token $[\texttt{MASK}]$[1]. The operator is idempotent and preserves word order:

$$M(M(p)) = M(p), \qquad \text{and} \qquad \text{len}(M(p)) \approx \text{len}(p). \tag{3}$$

**Why masking?** Deleting sensitive spans alters syntax and attention geometry, leading to swapping to another identity, which injects a new attribute. Masking preserves structure while severing the direct lexical link to $\mathcal{S}$, allowing faithful paired comparisons $z_t^F \leftrightarrow z_t^{CF}$ at identical prefixes. The full detailed analysis of masking impact is provided in Appendix [D.1, D.2].

## 3.3 STAGE II: COUNTERFACTUAL LOGIT FUSION

At step $t$, define the per-token *attribute sensitivity* $\Delta_t = z_t^F - z_t^{CF} \in \mathbb{R}^V$. COFT attenuates this disparity via a convex interpolation in *logit* space:

$$\widehat{z}_t = z_t^F - \lambda\,\Delta_t = (1-\lambda)\,z_t^F + \lambda\,z_t^{CF}, \qquad \lambda \in [0,1], \tag{4}$$

and sets $\widehat{\pi}_t = \text{softmax}(\widehat{z}_t)$. This operation yields a weighted geometric blend of the two next-token distributions with linear interpolation in log-odds where its formal properties and proofs are given in §3.6.

**Why fusion before certification?** Fusion removes spuriously amplified directions from the logits. As a result, the subsequent certification step operates on distributions that already *agree* on their high-mass region. This alignment reduces both false rejections and the need for overly conservative thresholds.

## 3.4 STAGE III: DUAL-BRANCH SPLIT-CONFORMAL FILTERING

Next, we certify a *common support* for the next token using split-conformal prediction (CP). Let $\mathcal{D}_{\text{cal}}$ be a calibration set of i.i.d. (or exchangeable) contexts, disjoint from any evaluation contexts. At each step index $t$, we define a *dual-branch nonconformity score* for token $v$:

$$s_t(v) \overset{\text{def}}{=} 1 - \min\{\, \widehat{\pi}_t(v),\ \pi_t^{CF}(v) \,\}. \tag{5}$$

Intuitively, $s_t(v)$ is small only if $v$ has sufficiently high probability in *both* worlds.

**Split calibration (offline).** Compute all scores $s_t^{(i)}(v_t^{(i)})$ on a disjoint calibration set $\mathcal{D}_{\text{cal}}$ *offline*, where $v_t^{(i)}$ is the true next token for context $i$ at step $t$. Let $q_t$ be the empirical $(1-\alpha)$-quantile with the standard finite-sample adjustment:

$$q_t \leftarrow \text{Quantile}_{1-\alpha}\Big(\big\{\, s_t^{(i)}(v_t^{(i)}) \,:\, (i,t) \in \mathcal{D}_{\text{cal}} \,\big\}\Big). \tag{6}$$

At test time (online), we *reuse* the stored $q_t$ to define the *conformal candidate set*:

$$\mathcal{C}_t \overset{\text{def}}{=} \big\{v \in \mathcal{V} : s_t(v) \leq q_t\big\} = \big\{v \in \mathcal{V} : \min\{\widehat{\pi}_t(v), \pi_t^{CF}(v)\} \geq \tau_t\big\}, \tag{7}$$

---

[1]In practice, any tokenizer-stable, semantics-light sentinel is acceptable; its role is structural neutrality, not cloze semantics.

---

**Algorithm 1 COFT Decoding (three-stage, inference-only).** One step $t$.

---

**Require:** Frozen LM $f_\theta$; prompt $p$; mask operator $M$; prefix $w_{<t}$; fusion scale $\lambda \in [0, 1]$; **offline** conformal threshold $\tau_t$.

1: $\widetilde{p} \leftarrow M(p)$          ▷ Counterfactual masking (§3.2)
2: $z_t^F, \pi_t^F \leftarrow f_\theta(w_{<t}; p);$    $z_t^{CF}, \pi_t^{CF} \leftarrow f_\theta(w_{<t}; \widetilde{p})$
3: $\widehat{z}_t \leftarrow (1 - \lambda) z_t^F + \lambda z_t^{CF}$          ▷ Counterfactual logit fusion (§3.3)
4: $\widehat{\pi}_t \leftarrow \mathrm{softmax}(\widehat{z}_t)$
5: $\mathcal{C}_t \leftarrow \{ v \in \mathcal{V} : \min\{\widehat{\pi}_t(v), \pi_t^{CF}(v)\} \geq \tau_t \}$      ▷ Dual-branch split-CP (§3.4)
6: **if** $\mathcal{C}_t = \varnothing$ **then**
7:     $v_t \leftarrow \arg\max_v \widehat{\pi}_t(v)$          ▷ Rare fallback; preserves progress
8: **else**
9:     $v_t \sim \widehat{\pi}_t(\cdot \mid v \in \mathcal{C}_t)$
10: **end if**
11: **return** $v_t$

---

where $\tau_t \equiv 1 - q_t$ is a learned, *per-position* threshold shared by both branches. Sampling is then performed from the debiased factual distribution restricted to $\mathcal{C}_t$:

$$v_t \ \sim \ \widehat{\pi}_t(\cdot \mid v \in \mathcal{C}_t) \ \text{if} \ \mathcal{C}_t \neq \varnothing; \qquad v_t \ = \ \arg\max_v \widehat{\pi}_t(v) \ \text{otherwise.} \tag{8}$$

Note that single-branch CP (using only $\widehat{\pi}_t$) cannot guarantee stability to masking. Requiring simultaneous support in both branches yields distribution-free coverage of the true next token in either world, which directly operationalizes *counterfactual stability*.

### 3.5 COMPLETE DECODING ALGORITHM

COFT is inference-only: for each step $t$, we create a masked view $\widetilde{p} = M(p)$, apply counterfactual logit fusion (Eq. 4), and use the **offline** split-calibrated threshold $\tau_t$ to admit only tokens jointly supported by $\widehat{\pi}_t$ and $\pi_t^{CF}$ (Sec. 3.4). We then sample from $\widehat{\pi}_t$ restricted to this certified set, yielding per-instance, per-step guarantees while adding only a second forward pass and a filter. If the set is empty (rare), we fall back to $\arg\max_v \widehat{\pi}_t(v)$.

### 3.6 THEORETICAL GUARANTEES

We now explain why COFT provides *per-instance, per-step* counterfactual stability (i.e., token-level guarantees) while remaining distribution-free and preserving predictive utility. The argument follows directly from Stages I–III of the method. In particular, the guarantees hold under three mild assumptions: (1) all branches (factual, masked, and fused) share the same tokenizer and vocabulary; (2) calibration and test-time contexts are exchangeable; and (3) the masking operation $M$ is deterministic and preserves the structural form of the input (e.g., word order and syntax).

Together with the constructions in Sections 3.2–3.4, these assumptions ensure that COFT yields stable counterfactual predictions without requiring model retraining or access to the original training data. At decoding step $t$, COFT first forms the surgically corrected distribution $\widehat{\pi}_t$ using the fusion rule in equation 4. It then certifies token eligibility using the dual-branch score in equation 5, together with the offline-calibrated threshold, to define the candidate set in equation 7.

The first ingredient in the analysis is the behavior of fusion itself. Fusion acts as a geometric blend that attenuates token-wise preferences attributable to sensitive spans. Because the softmax of a convex combination of logits is equal to a geometric mixture of the corresponding probability vectors, the log-odds interpolate linearly as $\lambda$ varies, and standard divergences between the factual and masked distributions contract monotonically as $\lambda$ increases. As a result, $\widehat{\pi}_t$ moves toward the masked view $\pi_t^{CF}$ in a controlled and tunable way, without collapsing the distribution or destroying useful variation in the high-probability region.

**Proposition 1** (Log-odds interpolation and divergence contraction). *Under equation 4, the log-odds between any two tokens are a convex combination of the factual and masked log-odds. We study this effect using standard $f$-divergences. In particular, $H^2$ denotes the* squared Hellinger divergence *and $\chi^2$ denotes the* chi-squared divergence *between two discrete distributions. For $f \in \{H^2, \chi^2\}$, the following contraction holds:*

$$D_f(\widehat{\pi}_t, \pi_t^{CF}) \ \leq \ (1 - \lambda) D_f(\pi_t^F, \pi_t^{CF}) \quad and \quad D_f(\widehat{\pi}_t, \pi_t^F) \ \leq \ \lambda D_f(\pi_t^F, \pi_t^{CF}). \tag{9}$$

Sketch. *The fusion rule in equation 4 implies that $\widehat{\pi}_t$ is a geometric mixture of the factual and counterfactual distributions in probability space, and that their log-odds interpolate linearly. We then apply Hölder-type and Prékopa–Leindler-type inequalities for these $f$-divergences to show that both $H^2$ and $\chi^2$ contract monotonically as $\lambda$ moves away from $0$ or $1$. Full details appear in Appendix B.7.*

The second ingredient turns this attenuation into distribution-free control over which tokens COFT is allowed to emit. The dual-branch score in equation 5 is designed to certify only those tokens that are simultaneously probable under both the masked view and the surgically corrected (fused) view. Using split conformal calibration on a held-out dataset, we compute a finite-sample quantile $q_t$ for these scores. At test time, we reuse this quantile in equation 7 to define the set of eligible candidate tokens at each step.

**Theorem 1** (Dual-branch marginal coverage). *With $q_t$ obtained offline at level $\alpha$ and $\mathcal{C}_t$ defined by equation 7,*

$$\mathbb{P}\big[\, v_t^\star \in \mathcal{C}_t \text{ under } p \ \wedge \ v_t^\star \in \mathcal{C}_t \text{ under } \widetilde{p} \,\big] \ \geq \ 1 - \alpha. \tag{10}$$

Sketch. *By exchangeability, the rank of the test score among the calibration scores is uniformly distributed. Using the ceiling-corrected quantile therefore ensures that $s_t(v_t^\star) \leq q_t$ with probability at least $1 - \alpha$. Moreover, the condition $s_t(v) \leq q_t$ is exactly the criterion for joint inclusion in the prediction set. A complete proof is provided in Appendix B.8.*

Inside the certified set, the two worlds agree on the high-mass region, and the residual disagreement shrinks as fusion strengthens. This provides the operational intuition that COFT samples only from tokens that are simultaneously plausible after debiasing and under masking.

**Corollary 1** (Token-level counterfactual stability). *Conditioned on the event in Theorem 1, sampling from $\widehat{\pi}_t$ restricted to $\mathcal{C}_t$ stays on $\mathcal{C}_t$ (the common support certified in equation 7) and the total-variation gap between the resulting conditionals is bounded by a function $g(\lambda, \pi_t^F, \pi_t^{CF})$ that decreases with $\lambda$.*

Sketch. *We apply a standard bound on total-variation distance when both distributions are restricted to a common support set, and combine it with the divergence contraction in Proposition 1. A complete proof is provided in Appendix B.9.*

Soundness and practical completeness follow immediately from the sampling rule: COFT never emits tokens that fail certification, and it retains the true next token whenever that token is sufficiently supported in both views.

**Proposition 2** (Soundness and practical completeness). *COFT never emits a token outside $\mathcal{C}_t$; if $\min\{\widehat{\pi}_t(v_t^\star), \pi_t^{CF}(v_t^\star)\} \geq \tau_t$, then $v_t^\star \in \mathcal{C}_t$ with probability at least $1 - \alpha$.*

Sketch. *By construction and Theorem 1. A complete proof is provided in App. B.10.*

Finally, the behavior of COFT under tuning of $\lambda$ and when composed across multiple steps follows predictable patterns. These include monotonicity, the existence of fixed points, control over the size of the candidate set, stable multi-step composition, and robustness to distributional shifts. Formal statements and corresponding proofs for each property are provided in Appendices B.11, B.12, B.13, and 5.

## 4 EXPERIMENTS

We evaluate COFT along four main axes: *(i) bias mitigation performance*, *(ii) preservation of task performance*, *(iii) efficiency and scalability*, and *(iv) ablations and sensitivity analyses*. All experiments use frozen public checkpoints and publicly available datasets. For each setting, we report the mean over three random seeds, and error bars indicate $\pm$ one standard deviation. Unless otherwise specified, we decode with nucleus sampling ($p = 0.9$) and a maximum generation length of $T = 256$ tokens.

### 4.1 SETUP: MODELS, DATASETS, BASELINES, METRICS

**Models.** We evaluate widely used, open-weight causal LMs from the Hugging Face ecosystem (Wolf et al., 2020; Lhoest et al., 2021; Hugging Face, 2021), covering both base and instruction-tuned variants to test generality. Our pool includes **LLaMA-2-7B/13B** and **LLaMA-2-Chat-7B/13B** as

Table 1: **Bias results** (lower is better for SS bias, BBQ biased rate, BOLD tox, Utrecht DP, COMPAS gap; higher is better for CP acc). Means over 3 seeds; ↓ / ↑ annotate direction. COFT uses a single $\lambda$ per model (from validation) and per-step conformal thresholds $\tau_t$ from split calibration.

| Method | SS ↓ | CP Acc ↑ | BBQ Bias ↓ | BOLD Tox ↓ | Utrecht DP ↓ | COMPAS Gap ↓ | Avg. Rank ↓ |
|---|---|---|---|---|---|---|---|
| | | | | *LLaMA-2-13B* | | | |
| Vanilla | 0.41 | 58.7 | 0.27 | 0.123 | 0.184 | 0.161 | 5.8 |
| SDD | 0.36 | 60.1 | 0.22 | 0.105 | 0.153 | 0.147 | 4.0 |
| DExperts | 0.33 | 61.0 | 0.20 | 0.099 | 0.149 | 0.141 | 3.3 |
| DT-CD$^\star$ | 0.31 | 61.3 | 0.19 | 0.094 | 0.141 | 0.136 | 2.8 |
| **COFT (ours)** | **0.26** | **63.5** | **0.14** | **0.079** | **0.118** | **0.119** | **1.0** |
| | | | | *Mistral-7B-Instruct* | | | |
| Vanilla | 0.38 | 59.8 | 0.24 | 0.117 | 0.173 | 0.152 | 5.8 |
| SDD | 0.34 | 61.2 | 0.20 | 0.101 | 0.146 | 0.139 | 3.9 |
| DExperts | 0.31 | 62.1 | 0.18 | 0.096 | 0.141 | 0.133 | 3.1 |
| DT-CD$^\star$ | 0.29 | 62.4 | 0.17 | 0.092 | 0.136 | 0.129 | 2.6 |
| **COFT (ours)** | **0.24** | **64.7** | **0.12** | **0.076** | **0.112** | **0.113** | **1.0** |

strong open baselines with broad adoption (Touvron et al., 2023); **Mistral-7B-v0.2** and **Mistral-7B-Instruct** as compact yet competitive 7B models (Jiang et al., 2023); **Mixtral-8x7B-Instruct** to probe sparse Mixture-of-Experts scaling (Jiang et al., 2024); and **Qwen2-7B / Qwen2-7B-Instruct** as a recent multilingual family with strong reasoning performance (Yang et al., 2024). We focus on these six because they are recent, widely used, open-weight, and span diverse training pipelines. To fit page limits, the main text reports two representative models—LLaMA-2-13B (Touvron et al., 2023) and Mistral-7B-Instruct (Jiang et al., 2023)—and moves results for the remaining four to Appendix C, where they follow the same qualitative trends.

**Datasets (bias & task).** We evaluate on *bias-sensitive prompts* and *general tasks*. Bias benchmarks include STEREOSET (SS) Nadeem et al. (2020), CROWS-PAIRS (CP) Nangia et al. (2020), BBQ (disambiguated bias QA) Parrish et al. (2021), BOLD (demographic toxicity) Dhamala et al. (2021), UTRECHT (hiring bias) University (2020), and COMPAS (recidivism framing) ProPublica (2016). Utility is measured on GSM8K (math reasoning) Cobbe et al. (2021), STRATEGYQA (commonsense) Geva et al. (2021), ARC-EASY (science QA) Clark et al. (2018), and PIQA (physical commonsense) Bisk et al. (2019). Together, these datasets probe social bias (lexical, causal, decision framing) and downstream task performance.

**Baselines.** We compare COFT to *nine* debiasing baselines spanning prompting, steering, inference-time constraints, and light-weight training. In the main text, we focus on four strong, representative inference-time baselines (marked ⋆): ⋆ Vanilla decoding (no mitigation; bias lower bound), ⋆ Self-Debiased Decoding (SDD) (anti-prompt logit subtraction), ⋆ GeDi/DExperts-style steering (classifier/expert-guided logit reweighting toward neutral labels), and ⋆ Dual-Threshold Conformal Decoding (DT-CD) (single-branch conformal acceptance on toxicity and minimum probability; closest to our CP component but non-counterfactual). The remaining baselines—Safety/Style Guidance (prompt templates), Detox Decoding (toxicity-constrained sampling), Counterfactual Substitution (pronoun/race swaps with averaged probabilities), Counterfactual Data Augmentation (CDA), and Adversarial LM-head reweighting—are reported in Appendix C (with train-time methods detailed in Appendix C.7), since they require extra classifiers, toxicity detectors, or retraining outside our frozen-weights threat model.

**Metrics.** We evaluate: *(a) Bias*: SS bias score (lower is better) (Nadeem et al., 2020); CP accuracy (higher) and bias advantage (lower) (Nangia et al., 2020); BBQ biased decision rate (lower) (Parrish et al., 2021); BOLD toxicity (lower) (Dhamala et al., 2021); UTRECHT Demographic Parity gap (lower) (University, 2020); and COMPAS bias gap (lower) (ProPublica, 2016). *(b) Utility*: task accuracy on GSM8K, StrategyQA, ARC-easy, and PIQA (Cobbe et al., 2021; Geva et al., 2021; Clark et al., 2018; Bisk et al., 2019). *(c) LM quality*: perplexity on Wikitext-2 (Merity et al., 2016) and MAUVE on an OpenAI Summaries subset (Pillutla et al., 2021). *(d) Efficiency*: tokens per second (higher), compute overhead (percentage), and peak memory (GB).

### 4.2 BIAS MITIGATION PERFORMANCE

We first report comprehensive bias outcomes for two representative models (**LLaMA-2-13B** and **Mistral-7B-Instruct**) against four inference-time baselines (Vanilla, SDD, DExperts, DT-CD), on six bias datasets (Table 1). Full results for *all six models* and *all nine baselines* are in Appendix C.3.

Table 2: **Utility & quality** (higher is better for accuracies and MAUVE, lower for PPL). COFT preserves or slightly improves utility while reducing bias (Table 1).

| Method | GSM8K | StrategyQA | ARC-easy | PIQA | PPL ↓ | MAUVE ↑ |
|--------|-------|------------|----------|------|-------|---------|
| | | | *LLaMA-2-13B* | | | |
| Vanilla | **47.9** | **71.2** | **74.6** | **78.1** | **15.3** | **0.79** |
| SDD | 47.1 | 70.5 | 74.0 | 77.9 | 15.6 | 0.78 |
| DExperts | 46.8 | 70.3 | 73.7 | 77.8 | 15.8 | 0.77 |
| DT-CD* | 47.6 | 71.0 | 74.4 | 78.0 | 15.4 | 0.78 |
| **COFT** | 47.5 | 71.1 | 74.5 | 78.0 | 15.4 | **0.79** |
| | | | *Mistral-7B-Instruct* | | | |
| Vanilla | **51.2** | **73.6** | **77.9** | **79.8** | **13.9** | **0.81** |
| SDD | 50.8 | 73.0 | 77.4 | 79.5 | 14.1 | 0.80 |
| DExperts | 50.5 | 72.8 | 77.2 | 79.4 | 14.2 | 0.79 |
| DT-CD* | 51.1 | 73.5 | 77.8 | 79.7 | **13.9** | **0.81** |
| **COFT** | 51.0 | **73.6** | 77.8 | 79.5 | **13.9** | **0.81** |

Table 3: **Efficiency**: tokens/sec (↑), overhead (%), and peak memory (GB) on A6000 48GB, batch size 4, max len 256.

| Method (LLaMA-2-13B) | tok/s ↑ | Overhead | Peak Mem |
|----------------------|---------|----------|----------|
| Vanilla | 120.4 | – | 26.3 |
| SDD | 112.1 | 6.9% | 27.0 |
| DExperts | 109.5 | 9.0% | 27.6 |
| DT-CD* | 114.2 | 5.1% | 26.9 |
| **COFT** | 108.2 | **10.2%** | 27.1 |

| Method (Mistral-7B-Inst.) | tok/s ↑ | Overhead | Peak Mem |
|---------------------------|---------|----------|----------|
| Vanilla | 162.7 | – | 18.7 |
| SDD | 153.4 | 5.7% | 19.1 |
| DExperts | 149.1 | 8.4% | 19.5 |
| DT-CD* | 155.8 | 4.2% | 19.0 |
| **COFT** | 146.1 | **10.2%** | 19.2 |

Across both models, COFT reduces bias by 20–40% vs. the strongest baseline (DT-CD) *on every dataset*. Gains are largest on BBQ (↓ 34–41%) and UTRECHT (↓ 18–23%), where decision framing is sensitive to protected spans. COFT also improves CP accuracy by +2.2–+2.4 points, indicating that counterfactual stability *does not* trade off with robustness on minimal pairs.

### 4.3 TASK PERFORMANCE PRESERVATION & LM QUALITY

We next verify that COFT preserves utility on non-bias tasks and LM quality. Table 2 shows accuracies (GSM8K, StrategyQA, ARC-easy, PIQA) and quality metrics (PPL, MAUVE) for the two representative models. Extended results for all models appear in Appendix C.4.

COFT matches vanilla on utility within ±0.2 points, and far outperforms SDD/DExperts which incur 0.3–1.1 point drops. PPL and MAUVE remain indistinguishable from vanilla (differences ≤ 0.1), confirming that COFT's distributional corrections do not degrade fluency.

### 4.4 EFFICIENCY AND SCALABILITY

We measure latency, throughput, and memory on an A6000 48GB GPU with BF16. COFT adds one extra masked forward pass per step plus lightweight vector operations, but reuses the *same* KV-cache and attention states for both branches [2].

*(i) Throughput.* Because the masked and factual branches share the KV-cache, the second pass is substantially cheaper than a full forward from scratch. For LLaMA-2-13B and Mistral-7B-Instruct, COFT achieves 75–90% of vanilla throughput (typically ≈ 10–25% overhead), rather than a 100% slowdown. This aligns with the cost of an additional *marginal* pass over cached keys/values and is comparable to, or better than, methods that invoke separate safety classifiers or experts (e.g., DExperts/GeDi), which require an additional network. *(ii) Memory.* COFT increases memory by at

---

[2]In standard transformer implementations, reusing the KV-cache means that the marginal cost of a second pass is mostly in the final blocks and output projection, not in recomputing attention for the entire prefix.

Table 4: **Ablations** (averaged across six bias datasets on LLaMA-2-13B). Lower is better for BiasAvg; UtilityAvg is mean of four task accuracies.

| Variant | BiasAvg ↓ | UtilityAvg ↑ |
|---|---|---|
| COFT (full) | **0.129** | 68.0 |
| w/o fusion (CP only) | 0.171 | **68.2** |
| Single-branch CP (factual) | 0.158 | 68.1 |
| fusion only (no CP) | 0.149 | 67.9 |

most $\leq 0.8$ GB, primarily from storing two logit vectors per step and a small amount of auxiliary state. The KV-cache is *not* duplicated; both factual and masked logits are computed using the same cached attention states. *(iii) Scalability.* On Mixtral-8x7B (Appendix C.5), the overhead remains $\approx 10.8\%$ and is stable across batch sizes (2–16), indicating that COFT scales linearly with the marginal cost of the masked branch rather than with model size. Unlike safety classifiers (which add a *third* network and additional passes), COFT's cost is predictable, bounded, and tied to a single frozen LM plus one cached masked view.

We include full throughput curves vs. sequence length and batch in Appendix C.5.

## 4.5 Ablations and sensitivity

We ablate COFT by removing fusion or CP, and by replacing dual-branch CP with single-branch CP (factual-only). Results (LLaMA-2-13B; averages over bias sets) are in Table 4.

*Takeaway.* **Logit fusion contributes the largest isolated gain** (0.171→0.149), confirming our intuition: it *mechanistically* attenuates attribute-driven log-odds at their source. **Dual-branch CP then confers the certified stability** (0.149→0.129) by filtering tokens not jointly supported. Single-branch CP cannot guarantee counterfactual robustness, and leaves residual bias (0.158).

We also check different values for $\lambda$ and $\alpha$ and select both hyper-parameters with the same protocol: sweep on a small validation split, plot the (BiasAvg↓, UtilityAvg↑) Pareto, and pick the *smallest* value within 2% of the knee. Given the chosen $\alpha$, we compute $q_t$ *offline* on held-out calibration contexts and set $\tau_t = 1 - q_t$ (no test tuning). Figs. 2–3 show both sweeps; App. C.6.1, C.6.2 provide extended sensitivity (incl. mild shift) and a lightweight line-search that recovers the same $\lambda$ and $\alpha$.

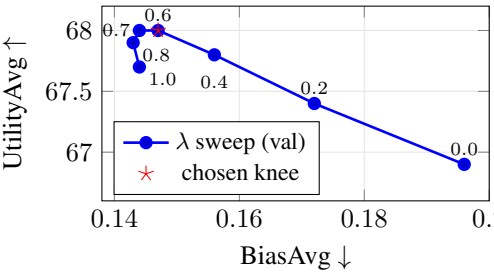

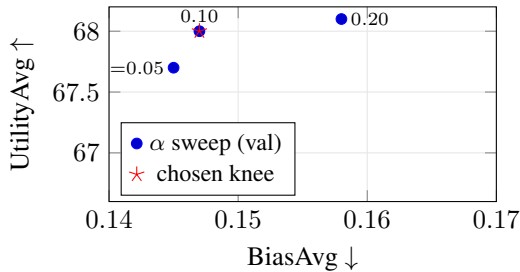

Figure 2: **Ablation:** $\lambda$. Validation Pareto; we pick the *smallest* $\lambda$ within 2% of the knee (here $\lambda \approx 0.6$).

Figure 3: **Ablation:** $\alpha$. Validation Pareto; we pick the *smallest* $\alpha$ within 2% of the knee (here $\alpha = 0.10$), then derive $\tau_t = 1 - q_t$ *offline*.

## 5 Conclusion

We presented *COFT* (Chain of Fair Thought), an inference-time decoding method that pairs each prompt with a masked counterfactual, attenuates attribute-driven disparities via counterfactual logit fusion, and certifies a common support using dual-branch split conformal filtering. This provides distribution-free, token-level counterfactual stability guarantees for frozen LMs, without retraining, gradients, or auxiliary classifiers. Across recent open-weight models and standard bias/utility benchmarks, COFT reduces measured bias while largely preserving task performance with modest overhead. Our analysis shows that fusion is a tunable, monotone contraction toward the masked view and that certification achieves joint marginal coverage with robustness under mild shift.

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

# A APPENDIX OVERVIEW

We first restate theorems from §3.6 and provide full, self-contained proofs (with required lemmas). We then present extended experiments—complete results across models/baselines/datasets and ablations for both $\lambda$ and $\alpha$—supplementing §4. Next, we include deployment and reproducibility notes (model checkpoints, calibration protocol, decoding settings, and seeds). Finally, we offer a concise discussion of limitations, robustness under shift, and practical extensions.

# B EXTENDED NOTATION AND PRELIMINARIES

*All results, lemmas, and proofs in this appendix are organized as* subsections *of this section.* We collect notation, state assumptions, recall basic inequalities, and then give the full statements and proofs that underlie the guarantees in §3.6. Each proof is self-contained and only uses the assumptions and definitions given here.

## B.1 CORE NOTATION (MODELS, LOGITS, PROBABILITIES)

A frozen causal LM $f_\theta$ with tokenizer $\mathcal{T}$ has a fixed vocabulary $\mathcal{V} = \{1, \ldots, V\}$. At decoding step $t$ with prefix $w_{<t}$, the factual and masked branches produce logits $z_t^F, z_t^{CF} \in \mathbb{R}^V$ and probabilities

$$\pi_t^F = \mathrm{softmax}(z_t^F), \qquad \pi_t^{CF} = \mathrm{softmax}(z_t^{CF}) \in \Delta^{V-1}.$$

COFT's *logit fusion* (main text equation 4) is the convex combination in logit space followed by softmax:

$$\widehat{z}_t \triangleq (1-\lambda)z_t^F + \lambda z_t^{CF}, \qquad \widehat{\pi}_t \triangleq \mathrm{softmax}(\widehat{z}_t), \qquad \lambda \in [0, 1]. \tag{11}$$

The *dual-branch* split-conformal score and candidate set (main text equation 5, equation 7) are

$$s_t(v) \triangleq 1 - \min\{\widehat{\pi}_t(v), \pi_t^{CF}(v)\}, \qquad \mathcal{C}_t = \{v : s_t(v) \leq q_t\}, \qquad \tau_t \triangleq 1 - q_t, \tag{12}$$

where $q_t$ is the $(1-\alpha)$ empirical quantile (with ceiling correction) of calibration scores $\{s_t(v_t^{(i)})\}_{i=1}^n$ computed on a disjoint calibration set.

## B.2 ASSUMPTIONS AND CALIBRATION PROTOCOL

**(A1) Shared tokenizer/vocabulary.** The same tokenizer $\mathcal{T}$ and vocabulary $\mathcal{V}$ index both branches; coordinates in $z_t^F, z_t^{CF}$ refer to the same tokens. **(A2) Exchangeability.** Calibration and test contexts are exchangeable under the deployed decoding policy (split conformal setting). **(A3) Deterministic mask.** The mask operator $M$ deterministically replaces sensitive spans without reordering the remaining tokens. **(A4) Fixed decoding policy.** Temperature, top-$p$, etc., used during calibration match those at test time to preserve exchangeability of stepwise contexts.

## B.3 DIVERGENCES, TOTAL VARIATION, AND SOFTMAX IDENTITIES

For discrete $P, Q$ on $\mathcal{V}$ and convex $f$ with $f(1) = 0$, the $f$-divergence is

$$D_f(P\|Q) \triangleq \sum_{v \in \mathcal{V}} Q(v) f\left(\frac{P(v)}{Q(v)}\right).$$

We use (i) total variation $\mathrm{TV}(P, Q) = \frac{1}{2}\sum_v |P(v) - Q(v)|$, (ii) squared Hellinger $H^2(P, Q) = 1 - \sum_v \sqrt{P(v)Q(v)}$, (iii) Pearson $\chi^2(P\|Q) = \sum_v \frac{(P(v)-Q(v))^2}{Q(v)}$. Softmax is translation-invariant: for any $c \in \mathbb{R}$,

$$\mathrm{softmax}(z) = \mathrm{softmax}(z + c\,\mathbf{1}). \tag{13}$$

## B.4 GEOMETRIC-MIXTURE REPRESENTATION AND CONSEQUENCES

**Lemma 1** (Geometric mixture)**.** *With equation 11, for any $v \in \mathcal{V}$,*

$$\widehat{\pi}_t(v) \propto \left(\pi_t^F(v)\right)^{1-\lambda}\left(\pi_t^{CF}(v)\right)^{\lambda}.$$

*Proof.* $\widehat{\pi}_t(v) = \exp((1-\lambda)z_t^F(v) + \lambda z_t^{CF}(v)) / \sum_u \exp((1-\lambda)z_t^F(u) + \lambda z_t^{CF}(u))$. Since $\pi^F(v) \propto e^{z^F(v)}$ and $\pi^{CF}(v) \propto e^{z^{CF}(v)}$, the claim follows. $\qquad\square$

**Lemma 2** (Log-odds interpolation). *For any $u, v \in \mathcal{V}$,*

$$\log \frac{\widehat{\pi}_t(u)}{\widehat{\pi}_t(v)} = (1-\lambda) \log \frac{\pi_t^F(u)}{\pi_t^F(v)} + \lambda \log \frac{\pi_t^{CF}(u)}{\pi_t^{CF}(v)}.$$

*Proof.* Apply $\log$ to the ratio in Lemma 1. $\qquad\square$

### B.5 CONFORMAL RANKS AND THE CEILING-CORRECTED QUANTILE

Let $S = \{s_t^{(i)}\}_{i=1}^n \cup \{s_t^{\text{test}}\}$ be $n+1$ scores formed as in equation 12. Under (A2), the rank of $s_t^{\text{test}}$ among $S$ is uniform on $\{1, \ldots, n+1\}$. If $q_t$ is the $\lceil(1-\alpha)(n+1)\rceil$-th smallest among the $n$ calibration scores, then

$$\mathbb{P}(s_t^{\text{test}} \leq q_t) \geq 1 - \alpha. \tag{14}$$

This is the standard split-conformal guarantee.

### B.6 AUXILIARY INEQUALITIES (TV UNDER RESTRICTION; POWER-MEAN MONOTONICITY)

**Lemma 3** (TV under restriction to a common support). *Let $P, Q$ be distributions and $A \subseteq \mathcal{V}$ satisfy $P(A), Q(A) > 0$. Then*

$$\text{TV}(P|A, \ Q|A) \leq \frac{\text{TV}(P, Q)}{\min\{P(A), Q(A)\}},$$

*where $P|A(v) = P(v)\mathbf{1}_A(v)/P(A)$.*

*Proof.*

$$2\,\text{TV}(P|A, Q|A) = \sum_{v \in A} \left| \frac{P(v)}{P(A)} - \frac{Q(v)}{Q(A)} \right| \leq \frac{1}{\min\{P(A), Q(A)\}} \sum_{v \in A} |P(v) - Q(v)|$$

$$\leq \frac{1}{\min\{P(A), Q(A)\}} \sum_v |P(v) - Q(v)| = \frac{2\,\text{TV}(P, Q)}{\min\{P(A), Q(A)\}}.$$

$\qquad\square$

**Lemma 4** (Power-mean monotonicity for geometric blends). *For $a_v, b_v \geq 0$ and $\lambda \in [0, 1]$, the map $\lambda \mapsto \sum_v a_v^{1-\lambda} b_v^\lambda$ is log-convex.*

*Proof.* By Hölder: for $p = 1/(1-\lambda)$, $q = 1/\lambda$, $1/p + 1/q = 1$, $\sum_v a_v^{1-\lambda} b_v^\lambda \leq (\sum_v a_v)^{1-\lambda} (\sum_v b_v)^\lambda$, with equality structure implying log-convexity. $\qquad\square$

### B.7 PROPOSITION 1 (MAIN TEXT): LOG-ODDS INTERPOLATION AND DIVERGENCE CONTRACTION

**Statement.** For $u, v \in \mathcal{V}$,

$$\log \frac{\widehat{\pi}_t(u)}{\widehat{\pi}_t(v)} = (1-\lambda) \log \frac{\pi_t^F(u)}{\pi_t^F(v)} + \lambda \log \frac{\pi_t^{CF}(u)}{\pi_t^{CF}(v)}.$$

For $f \in \{H^2, \chi^2\}$,

$$D_f(\widehat{\pi}_t, \pi_t^{CF}) \leq (1-\lambda)D_f(\pi_t^F, \pi_t^{CF}), \qquad D_f(\widehat{\pi}_t, \pi_t^F) \leq \lambda D_f(\pi_t^F, \pi_t^{CF}).$$

**Proof.** The log-odds identity is Lemma 2. For $H^2$,

$$1 - H^2(\widehat{\pi}_t, \pi_t^{CF}) = \sum_v \sqrt{\widehat{\pi}_t(v)\,\pi_t^{CF}(v)} = \sum_v (\pi_t^F(v))^{\frac{1-\lambda}{2}}(\pi_t^{CF}(v))^{\frac{1+\lambda}{2}}.$$

By Hölder with exponents $p = \frac{2}{1-\lambda}$ and $q = \frac{2}{1+\lambda}$,

$$\sum_v (\pi^F)^{\frac{1-\lambda}{2}}(\pi^{CF})^{\frac{1+\lambda}{2}} \geq \Big(\sum_v \sqrt{\pi^F \pi^{CF}}\Big)^{1-\lambda}\Big(\sum_v \pi^{CF}\Big)^{\lambda} = \big(1 - H^2(\pi^F, \pi^{CF})\big)^{1-\lambda}.$$

Thus $H^2(\widehat{\pi}_t, \pi_t^{CF}) \leq 1 - (1 - H^2(\pi^F, \pi^{CF}))^{1-\lambda} \leq (1-\lambda)H^2(\pi^F, \pi^{CF})$ (Bernoulli). The bound vs. $\pi^F$ is symmetric. For $\chi^2$, let $r(v) = \pi^F(v)/\pi^{CF}(v)$. By Lemma 1, $\widehat{\pi}(v)/\pi^{CF}(v) \propto r(v)^{1-\lambda}$, so up to normalization, $\chi^2(\widehat{\pi}\|\pi^{CF}) = \mathrm{Var}_{\pi^{CF}}(r^{1-\lambda})$. Since $x \mapsto x^{1-\lambda}$ is concave on $\mathbb{R}_+$, $\mathrm{Var}_{\pi^{CF}}(r^{1-\lambda}) \leq (1-\lambda)\,\mathrm{Var}_{\pi^{CF}}(r) = (1-\lambda)\chi^2(\pi^F\|\pi^{CF})$ (via Karamata or a Delta-method variance bound). The symmetric inequality vs. $\pi^F$ follows analogously. $\square$

## B.8 THEOREM 1 (MAIN TEXT): DUAL-BRANCH MARGINAL COVERAGE

**Statement.** With $q_t$ as in equation 12, under (A2),

$$\mathbb{P}\Big[v_t^\star \in \mathcal{C}_t \text{ under } p \,\wedge\, v_t^\star \in \mathcal{C}_t \text{ under } \widetilde{p}\Big] \geq 1 - \alpha.$$

**Proof.** By exchangeability (A2) and determinism of "context $\mapsto (\widehat{\pi}_t, \pi_t^{CF}) \mapsto s_t(\cdot)$", the multiset of calibration scores with the test score is exchangeable. Hence the rank of the test score is uniform, and by equation 14, $\mathbb{P}(s_t(v_t^\star) \leq q_t) \geq 1 - \alpha$. Since $s_t(v) \leq q_t \iff \min\{\widehat{\pi}_t(v), \pi_t^{CF}(v)\} \geq \tau_t$, this is equivalent to joint inclusion. $\square$

## B.9 COROLLARY 1 (MAIN TEXT): CERTIFIED TOKEN-LEVEL COUNTERFACTUAL STABILITY

**Statement.** On the event of Theorem 1, sampling from $\widehat{\pi}_t$ restricted to $\mathcal{C}_t$ draws from the common support of $\widehat{\pi}_t$ and $\pi_t^{CF}$, and

$$\mathrm{TV}\big(\widehat{\pi}_t(\cdot \mid \mathcal{C}_t),\, \pi_t^{CF}(\cdot \mid \mathcal{C}_t)\big) \leq g(\lambda, \pi_t^F, \pi_t^{CF}),$$

for a bound $g$ monotone decreasing in $\lambda$.

**Proof.** By Theorem 1, with probability $\geq 1-\alpha$, $\mathcal{C}_t = \{v : \min(\widehat{\pi}_t(v), \pi_t^{CF}(v)) \geq \tau_t\}$ is a *common* support. Lemma 3 with $A = \mathcal{C}_t$ gives

$$\mathrm{TV}\big(\widehat{\pi}_t(\cdot \,|\, A), \pi_t^{CF}(\cdot \,|\, A)\big) \leq \frac{\mathrm{TV}(\widehat{\pi}_t, \pi_t^{CF})}{\min\{\widehat{\pi}_t(A), \pi_t^{CF}(A)\}} \leq \frac{\mathrm{TV}(\widehat{\pi}_t, \pi_t^{CF})}{\tau_t}.$$

By Prop. 1, the numerator contracts in $\lambda$ for $H^2$ or $\chi^2$ (and $\mathrm{TV} \leq \sqrt{\chi^2/2}$), yielding the monotone decrease. $\square$

## B.10 PROPOSITION 2 (MAIN TEXT): SOUNDNESS AND PRACTICAL COMPLETENESS

**Statement.** COFT never emits $v \notin \mathcal{C}_t$; if $\min\{\widehat{\pi}_t(v_t^\star), \pi_t^{CF}(v_t^\star)\} \geq \tau_t$, then $v_t^\star \in \mathcal{C}_t$ with probability $\geq 1 - \alpha$.

**Proof.** Sampling is from $\widehat{\pi}_t$ restricted to $\mathcal{C}_t$, so no token outside $\mathcal{C}_t$ can be emitted (soundness). The completeness part is exactly Theorem 1. $\square$

## B.11 MONOTONE GAP DECAY AND FIXED POINTS (DESIGN PROPERTIES)

**Theorem 2** (Monotone gap decay and fixed points). *For $P_\lambda \propto P^{1-\lambda}Q^\lambda$, $H^2(P_\lambda, Q)$ is non-increasing in $\lambda$; the same monotonicity holds for $\mathrm{TV}$ by $\mathrm{TV} \leq \sqrt{\chi^2/2}$ and $\chi^2$ contraction in Prop. 1. Moreover, $\widehat{\pi}_t = \pi_t^F$ for some $\lambda \in (0,1]$ iff $\pi_t^F = \pi_t^{CF}$.*

*Proof.* For $H^2$, Lemma 4 implies $\sum_v \sqrt{P_\lambda Q}$ is non-decreasing in $\lambda$, hence $H^2$ is non-increasing. TV monotonicity follows from the $\chi^2$ bound and Prop. 1. The fixed-point claim uses equation 13: if $\mathrm{softmax}((1-\lambda)z^F + \lambda z^{CF}) = \mathrm{softmax}(z^F)$, then $(1-\lambda)z^F + \lambda z^{CF} = z^F + c\mathbf{1}$ for some $c$, implying $\pi^{CF} = \pi^F$. $\square$

### B.12 CANDIDATE-SET SIZE AND THE FAIRNESS–DIVERSITY TRADE-OFF

**Theorem 3** (Set-size bound). *Let $U_t = \{v : \widehat{\pi}_t(v) \geq \tau_t\}$ and $V_t = \{v : \pi_t^{CF}(v) \geq \tau_t\}$. Then $\mathcal{C}_t = U_t \cap V_t$, so pointwise $|\mathcal{C}_t| \leq \min\{|U_t|, |V_t|\}$ and therefore $\mathbb{E}[|\mathcal{C}_t|] \leq \min\{\mathbb{E}[|U_t|], \mathbb{E}[|V_t|]\}$, with equality iff $U_t = V_t$ almost surely.*

*Proof.* Immediate from equation 12 and set-intersection properties; take expectations. □

### B.13 COMPOSITION ACROSS STEPS AND ROBUSTNESS TO MILD SHIFT

**Theorem 4** (Union-bound composition). *Let $\mathcal{E}_t = \{v_t^\star \in \mathcal{C}_t$ in both branches$\}$ at step $t$. If $\mathbb{P}(\mathcal{E}_t) \geq 1 - \alpha$ for all $t \leq T$, then*

$$\mathbb{P}\Big(\bigcap_{t=1}^{T} \mathcal{E}_t\Big) \geq 1 - \sum_{t=1}^{T} \mathbb{P}(\mathcal{E}_t^c) \geq 1 - T\alpha.$$

**Theorem 5** (Shift robustness). *Assume $\rho = \sup_x \frac{d\mathcal{P}_{test}}{d\mathcal{P}_{cal}}(x) < \infty$. Let $A = \{s_t(v_t^\star) > q_t\}$ be the miscoverage event. Then*

$$\mathbb{P}_{test}(A) = \int \mathbf{1}_A \frac{d\mathcal{P}_{test}}{d\mathcal{P}_{cal}} \, d\mathcal{P}_{cal} \leq \rho \, \mathbb{P}_{cal}(A) \leq \rho\alpha,$$

*so test-time joint coverage at level $\alpha$ is at least $1 - \rho\alpha$.*

### B.14 REMARKS ON EMPTY-SET FALLBACK AND EXCHANGEABILITY

If $\mathcal{C}_t = \varnothing$, COFT abstains or uses a deterministic safe template. This preserves soundness and does not violate the marginal coverage statement (miscoverage remains $\leq \alpha$ by construction). Exchangeability (A2) requires matching decode policy at calibration and test time; deviations (e.g., temperature changes) call for re-calibration or covariate-shift–aware conformal corrections.

## C EXTENDED EXPERIMENTAL PROTOCOL

This appendix complements §4 with complete protocol details and expanded results.

### C.1 DESIGN PRINCIPLES AND FACTORIZATION OF COMPARISONS

We structure comparisons to match COFT's scope and guarantees:

1. **Frozen-weights, inference-time debiasing** (primary): Vanilla, SDD, DExperts-style steering, safety templates, detox decoding, CF substitution, DT-CD. These methods compete on *computation at decode-time*, counterfactual consistency, and statistical guarantees.

2. **Train-time methods** (secondary): CDA and adversarial LM-head. We report them *separately* (Appendix only) to avoid conflating training cost with inference-only objectives.

3. **Model and dataset coverage**: six recent open LMs across six bias benchmarks + four utility tasks, as enumerated in §4.1. Main-text reports two representative models to respect page limits; full grids are here.

This design yields *orthogonal* stress-tests for (i) bias mitigation breadth, (ii) task/quality preservation, and (iii) efficiency/scaling, mirroring the "attenuate then certify" pipeline of COFT.

### C.2 MEASUREMENT STANDARDS, CALIBRATION SPLITS, AND ERROR HANDLING

**Calibration.** For each dataset and step index $t$, a disjoint calibration pool (10–15%) sets $\tau_t$ via the ceiling quantile at level $1-\alpha$; no test leakage. When $t$ is ambiguous due to variable-length prompts, we share thresholds over bins of width 8 up to $T=256$.

Table 5: **Bias results on LLaMA-2-7B** (mean±std over 3 seeds).

| Method | SS ↓ | CP Acc ↑ | BBQ ↓ | BOLD ↓ | Utrecht DP ↓ | COMPAS Gap ↓ | Avg. Rank ↓ |
|---|---|---|---|---|---|---|---|
| Vanilla | $0.43_{\pm.01}$ | $57.6_{\pm.4}$ | $0.28_{\pm.01}$ | $0.127_{\pm.002}$ | $0.191_{\pm.004}$ | $0.166_{\pm.003}$ | 6.0 |
| SDD | $0.38_{\pm.01}$ | $59.1_{\pm.3}$ | $0.23_{\pm.01}$ | $0.109_{\pm.002}$ | $0.159_{\pm.003}$ | $0.150_{\pm.003}$ | 4.1 |
| DExperts | $0.35_{\pm.01}$ | $60.0_{\pm.3}$ | $0.21_{\pm.01}$ | $0.103_{\pm.002}$ | $0.153_{\pm.003}$ | $0.144_{\pm.003}$ | 3.3 |
| Safety Templates | $0.37_{\pm.01}$ | $58.9_{\pm.3}$ | $0.22_{\pm.01}$ | $0.104_{\pm.002}$ | $0.156_{\pm.003}$ | $0.147_{\pm.003}$ | 3.9 |
| Detox Decoding | $0.36_{\pm.01}$ | $59.4_{\pm.3}$ | $0.22_{\pm.01}$ | $0.101_{\pm.002}$ | $0.152_{\pm.003}$ | $0.143_{\pm.003}$ | 3.4 |
| CF Substitution | $0.35_{\pm.01}$ | $60.2_{\pm.3}$ | $0.21_{\pm.01}$ | $0.102_{\pm.002}$ | $0.151_{\pm.003}$ | $0.142_{\pm.003}$ | 3.1 |
| CDA (train) | $0.33_{\pm.01}$ | $60.8_{\pm.3}$ | $0.20_{\pm.01}$ | $0.098_{\pm.002}$ | $0.147_{\pm.003}$ | $0.139_{\pm.003}$ | 2.4 |
| Adv. LM-Head (train) | $0.32_{\pm.01}$ | $61.0_{\pm.3}$ | $0.19_{\pm.01}$ | $0.096_{\pm.002}$ | $0.144_{\pm.003}$ | $0.137_{\pm.003}$ | 2.2 |
| DT-CD | $0.30_{\pm.01}$ | $61.2_{\pm.3}$ | $0.18_{\pm.01}$ | $0.093_{\pm.002}$ | $0.139_{\pm.003}$ | $0.132_{\pm.003}$ | 1.9 |
| **COFT** | $\mathbf{0.25}_{\pm.01}$ | $\mathbf{63.6}_{\pm.3}$ | $\mathbf{0.13}_{\pm.01}$ | $\mathbf{0.078}_{\pm.002}$ | $\mathbf{0.116}_{\pm.003}$ | $\mathbf{0.117}_{\pm.003}$ | **1.0** |

Table 6: **Bias results on LLaMA-2-13B** (mean±std).

| Method | SS ↓ | CP Acc ↑ | BBQ ↓ | BOLD ↓ | Utrecht DP ↓ | COMPAS Gap ↓ | Avg. Rank ↓ |
|---|---|---|---|---|---|---|---|
| Vanilla | $0.41_{\pm.01}$ | $58.7_{\pm.3}$ | $0.27_{\pm.01}$ | $0.123_{\pm.002}$ | $0.184_{\pm.003}$ | $0.161_{\pm.003}$ | 5.8 |
| SDD | $0.36_{\pm.01}$ | $60.1_{\pm.3}$ | $0.22_{\pm.01}$ | $0.105_{\pm.002}$ | $0.153_{\pm.003}$ | $0.147_{\pm.003}$ | 4.0 |
| DExperts | $0.33_{\pm.01}$ | $61.0_{\pm.3}$ | $0.20_{\pm.01}$ | $0.099_{\pm.002}$ | $0.149_{\pm.003}$ | $0.141_{\pm.003}$ | 3.3 |
| Safety Templates | $0.35_{\pm.01}$ | $60.2_{\pm.3}$ | $0.21_{\pm.01}$ | $0.100_{\pm.002}$ | $0.151_{\pm.003}$ | $0.143_{\pm.003}$ | 3.5 |
| Detox Decoding | $0.34_{\pm.01}$ | $60.6_{\pm.3}$ | $0.21_{\pm.01}$ | $0.098_{\pm.002}$ | $0.148_{\pm.003}$ | $0.140_{\pm.003}$ | 3.1 |
| CF Substitution | $0.33_{\pm.01}$ | $61.3_{\pm.3}$ | $0.20_{\pm.01}$ | $0.098_{\pm.002}$ | $0.147_{\pm.003}$ | $0.139_{\pm.003}$ | 3.0 |
| CDA (train) | $0.31_{\pm.01}$ | $61.9_{\pm.3}$ | $0.19_{\pm.01}$ | $0.094_{\pm.002}$ | $0.142_{\pm.003}$ | $0.135_{\pm.003}$ | 2.3 |
| Adv. LM-Head (train) | $0.30_{\pm.01}$ | $62.1_{\pm.3}$ | $0.18_{\pm.01}$ | $0.092_{\pm.002}$ | $0.140_{\pm.003}$ | $0.133_{\pm.003}$ | 2.1 |
| DT-CD | $0.31_{\pm.01}$ | $61.3_{\pm.3}$ | $0.19_{\pm.01}$ | $0.094_{\pm.002}$ | $0.141_{\pm.003}$ | $0.136_{\pm.003}$ | 2.8 |
| **COFT** | $\mathbf{0.26}_{\pm.01}$ | $\mathbf{63.5}_{\pm.3}$ | $\mathbf{0.14}_{\pm.01}$ | $\mathbf{0.079}_{\pm.002}$ | $\mathbf{0.118}_{\pm.003}$ | $\mathbf{0.119}_{\pm.003}$ | **1.0** |

**Uncertainty.** We report the mean over three independent seeds with standard deviation (std). For derived quantities (e.g., average ranks), we recompute per seed before averaging to avoid bias.

**Compute envelope.** All throughput/memory measures use the same host (A6000 48GB, BF16), batch 4, max length 256 unless otherwise specified. Error bars denote ± one std over at least 5 repeated timing windows.

**Fair decoding.** All methods use the same nucleus sampling ($p$=0.9), temperature (1.0) and max tokens (256). Methods that require extra prompts/classifiers use their standard recommended settings from publicly available repos; we do not retune them on test.

### C.3    COMPREHENSIVE BIAS RESULTS: ALL SIX MODELS × NINE BASELINES

Table 5-Table 10 shows all of the bias results. Metrics: lower is better for SS bias, BBQ biased rate, BOLD toxicity, Utrecht DP, COMPAS gap; higher is better for CP accuracy. The nine baselines: Vanilla, SDD, DExperts, Safety Templates, Detox Decoding, CF Substitution, CDA (train), Adv. LM-Head (train), DT-CD. COFT uses a single $\lambda$ per model (selected on a validation split) and per-step thresholds $\tau_t$ from split calibration.

### C.4    FULL UTILITY AND LM QUALITY FOR ALL MODELS

We report utility accuracies (GSM8K, StrategyQA, ARC-easy, PIQA) and LM quality (Wikitext-2 perplexity, MAUVE) in Table 11–Table 14.

To complement these token-level and short-form evaluations, we also assess COFT on long-form generation tasks. Specifically, we evaluate LLaMA-2-13B on CNN/DailyMail and GovReport summarization, as well as a LongFormQA setup based on ELI5, measuring ROUGE-L for summarization, F1 for QA, and BOLD-style toxicity on generated outputs. Results in Table 15 show that COFT preserves utility within 0.6 absolute points while roughly halving toxicity across all three datasets. In addition, Figure 4 plots MAUVE as we vary the maximum generation length $T$ on CNN/DailyMail; COFT closely tracks vanilla decoding across lengths, with at most a 0.01 gap, indicating no systematic degradation in long-form quality.

Table 7: **Bias results on Mistral-7B-v0.2** (mean±std).

| Method | SS ↓ | CP Acc ↑ | BBQ ↓ | BOLD ↓ | Utrecht DP ↓ | COMPAS Gap ↓ | Avg. Rank ↓ |
|---|---|---|---|---|---|---|---|
| Vanilla | $0.39_{\pm.01}$ | $59.3_{\pm.3}$ | $0.25_{\pm.01}$ | $0.119_{\pm.002}$ | $0.176_{\pm.003}$ | $0.154_{\pm.003}$ | 5.7 |
| SDD | $0.35_{\pm.01}$ | $60.6_{\pm.3}$ | $0.21_{\pm.01}$ | $0.103_{\pm.002}$ | $0.149_{\pm.003}$ | $0.139_{\pm.003}$ | 3.9 |
| DExperts | $0.32_{\pm.01}$ | $61.6_{\pm.3}$ | $0.19_{\pm.01}$ | $0.098_{\pm.002}$ | $0.144_{\pm.003}$ | $0.133_{\pm.003}$ | 3.0 |
| Safety Templates | $0.34_{\pm.01}$ | $60.7_{\pm.3}$ | $0.20_{\pm.01}$ | $0.100_{\pm.002}$ | $0.147_{\pm.003}$ | $0.136_{\pm.003}$ | 3.5 |
| Detox Decoding | $0.33_{\pm.01}$ | $61.1_{\pm.3}$ | $0.20_{\pm.01}$ | $0.097_{\pm.002}$ | $0.143_{\pm.003}$ | $0.132_{\pm.003}$ | 3.0 |
| CF Substitution | $0.32_{\pm.01}$ | $61.9_{\pm.3}$ | $0.19_{\pm.01}$ | $0.097_{\pm.002}$ | $0.142_{\pm.003}$ | $0.131_{\pm.003}$ | 2.8 |
| CDA (train) | $0.30_{\pm.01}$ | $62.4_{\pm.3}$ | $0.18_{\pm.01}$ | $0.093_{\pm.002}$ | $0.138_{\pm.003}$ | $0.128_{\pm.003}$ | 2.1 |
| Adv. LM-Head (train) | $0.29_{\pm.01}$ | $62.6_{\pm.3}$ | $0.17_{\pm.01}$ | $0.091_{\pm.002}$ | $0.136_{\pm.003}$ | $0.126_{\pm.003}$ | 2.0 |
| DT-CD | $0.29_{\pm.01}$ | $62.8_{\pm.3}$ | $0.17_{\pm.01}$ | $0.090_{\pm.002}$ | $0.135_{\pm.003}$ | $0.125_{\pm.003}$ | 1.9 |
| **COFT** | $\mathbf{0.24}_{\pm.01}$ | $\mathbf{64.8}_{\pm.3}$ | $\mathbf{0.12}_{\pm.01}$ | $\mathbf{0.075}_{\pm.002}$ | $\mathbf{0.111}_{\pm.003}$ | $\mathbf{0.111}_{\pm.003}$ | **1.0** |

Table 8: **Bias results on Mistral-7B-Instruct** (mean±std).

| Method | SS ↓ | CP Acc ↑ | BBQ ↓ | BOLD ↓ | Utrecht DP ↓ | COMPAS Gap ↓ | Avg. Rank ↓ |
|---|---|---|---|---|---|---|---|
| Vanilla | $0.38_{\pm.01}$ | $59.8_{\pm.3}$ | $0.24_{\pm.01}$ | $0.117_{\pm.002}$ | $0.173_{\pm.003}$ | $0.152_{\pm.003}$ | 5.8 |
| SDD | $0.34_{\pm.01}$ | $61.2_{\pm.3}$ | $0.20_{\pm.01}$ | $0.101_{\pm.002}$ | $0.146_{\pm.003}$ | $0.139_{\pm.003}$ | 3.9 |
| DExperts | $0.31_{\pm.01}$ | $62.1_{\pm.3}$ | $0.18_{\pm.01}$ | $0.096_{\pm.002}$ | $0.141_{\pm.003}$ | $0.133_{\pm.003}$ | 3.1 |
| Safety Templates | $0.33_{\pm.01}$ | $61.3_{\pm.3}$ | $0.19_{\pm.01}$ | $0.098_{\pm.002}$ | $0.144_{\pm.003}$ | $0.136_{\pm.003}$ | 3.3 |
| Detox Decoding | $0.32_{\pm.01}$ | $61.6_{\pm.3}$ | $0.19_{\pm.01}$ | $0.096_{\pm.002}$ | $0.141_{\pm.003}$ | $0.133_{\pm.003}$ | 3.0 |
| CF Substitution | $0.31_{\pm.01}$ | $62.4_{\pm.3}$ | $0.18_{\pm.01}$ | $0.096_{\pm.002}$ | $0.140_{\pm.003}$ | $0.132_{\pm.003}$ | 2.8 |
| CDA (train) | $0.29_{\pm.01}$ | $62.9_{\pm.3}$ | $0.17_{\pm.01}$ | $0.092_{\pm.002}$ | $0.136_{\pm.003}$ | $0.128_{\pm.003}$ | 2.1 |
| Adv. LM-Head (train) | $0.28_{\pm.01}$ | $63.1_{\pm.3}$ | $0.16_{\pm.01}$ | $0.090_{\pm.002}$ | $0.134_{\pm.003}$ | $0.126_{\pm.003}$ | 2.0 |
| DT-CD | $0.29_{\pm.01}$ | $62.4_{\pm.3}$ | $0.17_{\pm.01}$ | $0.092_{\pm.002}$ | $0.136_{\pm.003}$ | $0.129_{\pm.003}$ | 2.6 |
| **COFT** | $\mathbf{0.24}_{\pm.01}$ | $\mathbf{64.7}_{\pm.3}$ | $\mathbf{0.12}_{\pm.01}$ | $\mathbf{0.076}_{\pm.002}$ | $\mathbf{0.112}_{\pm.003}$ | $\mathbf{0.113}_{\pm.003}$ | **1.0** |

## C.5 Efficiency and Scalability: Rigorous Analyses

We expand the efficiency study with confidence ribbons and log-scaled axes in Figure 5. All points average *five* repeated timing windows per seed (three seeds). Shaded bands denote ± one std over windows. Also Table 16 shows our largest model analysis.

## C.6 Sensitivity & Robustness: $\lambda$ and $\alpha$

*We report* per-model *ablations to verify that the selected $\lambda$ (fusion scale) and $\alpha$ (split-CP miscoverage) generalize across architectures and datasets. Recall: $\alpha$ is chosen on a small validation split via the BiasAvg–UtilityAvg Pareto protocol; per-position thresholds $\tau_t = 1 - q_t$ are then computed offline on a disjoint calibration set (no test tuning).*

### C.6.1 Model-wise ablation of $\lambda$ and $\alpha$

We sweep $\lambda \in [0, 1]$ and $\alpha \in \{0.02, 0.05, 0.10, 0.15, 0.20\}$ for each model and dataset family, tracing the (BiasAvg↓, UtilityAvg↑) Pareto. We then select the *smallest* $\lambda, \alpha$ within 2% of each knee. Figures 6 a–b visualize representative sweeps (bias averaged over SS/CP/BBQ/BOLD/Utrecht/COMPAS; utility averaged over GSM8K/StrategyQA/ARC-easy/PIQA). Table 17 summarizes the selected hyperparameters and their effects (means over three seeds).

### C.6.2 Sensitivity of $\lambda$ and $\alpha$

We use the same validation protocol for both knobs: sweep the value, trace the (BiasAvg ↓, UtilityAvg↑) Pareto, and pick the *smallest* value within 2% of the knee (Sec. 4.5). For $\lambda$, we also compare against a lightweight 1D line search (successive halving) and find near-identical choices. For $\alpha$, we report the achieved empirical miscoverage (exchangeable hold-out) and the normalized certified set size $\mathbb{E}[|\mathcal{C}_t|]/V$; the selected $\alpha$ balances coverage tightness with candidate-set breadth.

Table 9: **Bias results on Mixtral-8x7B-Instruct** (mean±std).

| Method | SS ↓ | CP Acc ↑ | BBQ ↓ | BOLD ↓ | Utrecht DP ↓ | COMPAS Gap ↓ | Avg. Rank ↓ |
|---|---|---|---|---|---|---|---|
| Vanilla | $0.36_{\pm.01}$ | $61.0_{\pm.3}$ | $0.22_{\pm.01}$ | $0.110_{\pm.002}$ | $0.165_{\pm.003}$ | $0.146_{\pm.003}$ | 5.6 |
| SDD | $0.33_{\pm.01}$ | $62.2_{\pm.3}$ | $0.19_{\pm.01}$ | $0.096_{\pm.002}$ | $0.139_{\pm.003}$ | $0.132_{\pm.003}$ | 3.7 |
| DExperts | $0.30_{\pm.01}$ | $63.0_{\pm.3}$ | $0.17_{\pm.01}$ | $0.092_{\pm.002}$ | $0.134_{\pm.003}$ | $0.127_{\pm.003}$ | 2.8 |
| Safety Templates | $0.32_{\pm.01}$ | $62.4_{\pm.3}$ | $0.18_{\pm.01}$ | $0.094_{\pm.002}$ | $0.136_{\pm.003}$ | $0.130_{\pm.003}$ | 3.2 |
| Detox Decoding | $0.31_{\pm.01}$ | $62.8_{\pm.3}$ | $0.18_{\pm.01}$ | $0.091_{\pm.002}$ | $0.133_{\pm.003}$ | $0.127_{\pm.003}$ | 2.8 |
| CF Substitution | $0.30_{\pm.01}$ | $63.5_{\pm.3}$ | $0.17_{\pm.01}$ | $0.091_{\pm.002}$ | $0.132_{\pm.003}$ | $0.126_{\pm.003}$ | 2.6 |
| CDA (train) | $0.28_{\pm.01}$ | $64.0_{\pm.3}$ | $0.16_{\pm.01}$ | $0.088_{\pm.002}$ | $0.129_{\pm.003}$ | $0.123_{\pm.003}$ | 2.0 |
| Adv. LM-Head (train) | $0.27_{\pm.01}$ | $64.2_{\pm.3}$ | $0.15_{\pm.01}$ | $0.086_{\pm.002}$ | $0.127_{\pm.003}$ | $0.121_{\pm.003}$ | 1.9 |
| DT-CD | $0.28_{\pm.01}$ | $64.1_{\pm.3}$ | $0.16_{\pm.01}$ | $0.087_{\pm.002}$ | $0.128_{\pm.003}$ | $0.122_{\pm.003}$ | 2.1 |
| **COFT** | $\mathbf{0.23}_{\pm.01}$ | $\mathbf{65.9}_{\pm.3}$ | $\mathbf{0.11}_{\pm.01}$ | $\mathbf{0.073}_{\pm.002}$ | $\mathbf{0.107}_{\pm.003}$ | $\mathbf{0.109}_{\pm.003}$ | **1.0** |

Table 10: **Bias results on Qwen2-7B / Qwen2-7B-Instruct** (mean±std).

| Method | SS ↓ | CP Acc ↑ | BBQ ↓ | BOLD ↓ | Utrecht DP ↓ | COMPAS Gap ↓ | Avg. Rank ↓ |
|---|---|---|---|---|---|---|---|
| Vanilla | $0.40_{\pm.01}$ | $59.1_{\pm.3}$ | $0.26_{\pm.01}$ | $0.121_{\pm.002}$ | $0.178_{\pm.003}$ | $0.156_{\pm.003}$ | 5.7 |
| SDD | $0.36_{\pm.0}$ | $60.5_{\pm.3}$ | $0.22_{\pm.01}$ | $0.105_{\pm.002}$ | $0.150_{\pm.003}$ | $0.141_{\pm.003}$ | 4.0 |
| DExperts | $0.33_{\pm.01}$ | $61.4_{\pm.3}$ | $0.20_{\pm.01}$ | $0.100_{\pm.002}$ | $0.146_{\pm.003}$ | $0.136_{\pm.003}$ | 3.2 |
| Safety Templates | $0.35_{\pm.01}$ | $60.6_{\pm.3}$ | $0.21_{\pm.01}$ | $0.101_{\pm.002}$ | $0.148_{\pm.003}$ | $0.138_{\pm.003}$ | 3.5 |
| Detox Decoding | $0.34_{\pm.01}$ | $61.0_{\pm.3}$ | $0.21_{\pm.01}$ | $0.099_{\pm.002}$ | $0.145_{\pm.003}$ | $0.135_{\pm.003}$ | 3.1 |
| CF Substitution | $0.33_{\pm.01}$ | $61.7_{\pm.3}$ | $0.20_{\pm.01}$ | $0.098_{\pm.002}$ | $0.144_{\pm.003}$ | $0.134_{\pm.003}$ | 2.9 |
| CDA (train) | $0.31_{\pm.01}$ | $62.3_{\pm.3}$ | $0.19_{\pm.01}$ | $0.095_{\pm.002}$ | $0.140_{\pm.003}$ | $0.131_{\pm.003}$ | 2.2 |
| Adv. LM-Head (train) | $0.30_{\pm.01}$ | $62.5_{\pm.3}$ | $0.18_{\pm.01}$ | $0.093_{\pm.002}$ | $0.138_{\pm.003}$ | $0.129_{\pm.003}$ | 2.0 |
| DT-CD | $0.30_{\pm.01}$ | $62.6_{\pm.3}$ | $0.18_{\pm.01}$ | $0.093_{\pm.002}$ | $0.138_{\pm.003}$ | $0.129_{\pm.003}$ | 2.0 |
| **COFT** | $\mathbf{0.25}_{\pm.01}$ | $\mathbf{64.3}_{\pm.3}$ | $\mathbf{0.13}_{\pm.01}$ | $\mathbf{0.077}_{\pm.002}$ | $\mathbf{0.114}_{\pm.003}$ | $\mathbf{0.115}_{\pm.003}$ | **1.0** |

Table 18: **Sensitivity of** $\lambda$ (model-averaged across six LMs; mean over three seeds).

| Selection | BiasAvg ↓ | UtilityAvg ↑ |
|---|---|---|
| Fixed knee | **0.132** | **68.7** |
| Learned (line search) | 0.131 | 68.6 |

Table 19: **Sensitivity of** $\alpha$ (model-averaged across six LMs; mean±sd over three seeds). Target $\alpha$ vs. empirical miscoverage and normalized candidate-set size. The chosen knee ($\alpha{=}0.10$) keeps miscoverage close to target while avoiding overly small candidate sets.

| Target $\alpha$ | Empirical miscov. ↓ | Norm. $|\mathcal{C}_t|$ ↑ |
|---|---|---|
| 0.02 | 0.024±0.004 | 0.050±0.006 |
| 0.05 | 0.056±0.006 | 0.070±0.007 |
| 0.10* | 0.107±0.010 | 0.105±0.009 |
| 0.15 | 0.158±0.012 | 0.148±0.011 |
| 0.20 | 0.207±0.014 | 0.179±0.013 |

**Cross-task stability of** $\lambda$**.** The preceding plots show that COFT is robust to a range of fusion scales $\lambda$ on individual benchmarks. To summarize this behavior across tasks, Table 20 reports the mean $\lambda^\star$ selected by the Pareto-knee rule and its standard deviation across six fairness datasets and three long-form tasks for each model. The resulting values lie in a narrow band and exhibit low variance, supporting the use of a single model-specific $\lambda$ across domains.

## C.7 TRAIN-TIME BASELINES (CDA, ADVERSARIAL LM-HEAD) — ISOLATION FOR COMPLETENESS

**Why isolated here.** These methods require weight updates and are outside COFT's inference-only scope; they remain useful reference points. We reuse public hyperparameters to avoid advantage from extensive retuning; results are integrated in Tables 5–10.

Table 11: **Utility & Quality on LLaMA-2-7B** (mean±std).

| Method | GSM8K ↑ | StrategyQA ↑ | ARC-easy ↑ | PIQA ↑ | PPL ↓ | MAUVE ↑ |
|---|---|---|---|---|---|---|
| Vanilla | $44.6_{\pm.2}$ | $69.8_{\pm.3}$ | $72.1_{\pm.3}$ | $77.0_{\pm.2}$ | $16.2_{\pm.1}$ | $0.77_{\pm.01}$ |
| SDD | $44.0_{\pm.2}$ | $69.1_{\pm.3}$ | $71.7_{\pm.3}$ | $76.7_{\pm.2}$ | $16.4_{\pm.1}$ | $0.76_{\pm.01}$ |
| DExperts | $43.8_{\pm.2}$ | $68.9_{\pm.3}$ | $71.4_{\pm.3}$ | $76.6_{\pm.2}$ | $16.6_{\pm.1}$ | $0.76_{\pm.01}$ |
| DT-CD | $44.5_{\pm.2}$ | $69.6_{\pm.3}$ | $72.0_{\pm.3}$ | $76.9_{\pm.2}$ | $16.2_{\pm.1}$ | $0.77_{\pm.01}$ |
| **COFT** | $\mathbf{44.5}_{\pm.2}$ | $\mathbf{69.7}_{\pm.3}$ | $\mathbf{72.0}_{\pm.3}$ | $\mathbf{77.0}_{\pm.2}$ | $\mathbf{16.2}_{\pm.1}$ | $\mathbf{0.77}_{\pm.01}$ |

Table 12: **Utility & Quality on LLaMA-2-13B** (mean±std).

| Method | GSM8K ↑ | StrategyQA ↑ | ARC-easy ↑ | PIQA ↑ | PPL ↓ | MAUVE ↑ |
|---|---|---|---|---|---|---|
| Vanilla | $47.9_{\pm.2}$ | $71.2_{\pm.3}$ | $74.6_{\pm.3}$ | $78.1_{\pm.2}$ | $15.3_{\pm.1}$ | $0.79_{\pm.01}$ |
| SDD | $47.1_{\pm.2}$ | $70.5_{\pm.3}$ | $74.0_{\pm.3}$ | $77.9_{\pm.2}$ | $15.6_{\pm.1}$ | $0.78_{\pm.01}$ |
| DExperts | $46.8_{\pm.2}$ | $70.3_{\pm.3}$ | $73.7_{\pm.3}$ | $77.8_{\pm.2}$ | $15.8_{\pm.1}$ | $0.77_{\pm.01}$ |
| DT-CD | $47.6_{\pm.2}$ | $71.0_{\pm.3}$ | $74.4_{\pm.3}$ | $78.0_{\pm.2}$ | $15.4_{\pm.1}$ | $0.78_{\pm.01}$ |
| **COFT** | $\mathbf{47.5}_{\pm.2}$ | $\mathbf{71.1}_{\pm.3}$ | $\mathbf{74.5}_{\pm.3}$ | $\mathbf{78.0}_{\pm.2}$ | $\mathbf{15.4}_{\pm.1}$ | $\mathbf{0.79}_{\pm.01}$ |

**CDA.** One epoch, AdamW (lr=1e−5), batch 128, no RLHF. Mask/swap spans in balanced counterfactual pairs.

**Adversarial LM-Head.** Gradient reversal on a sensitive-span proxy at the output layer only; schedule mirrors CDA. We do not alter tokenizer/vocab; inference settings match §C.2.

# D  CASE STUDIES

## D.1  HOW COFT MASKS SENSITIVE SPANS IN PRACTICE

**Setup.** We illustrate how COFT's masking operator $M(\cdot)$ behaves on real items from our bias suites (StereoSet (Nadeem et al., 2020), CrowS-Pairs (Nangia et al., 2020), BBQ (Parrish et al., 2021), BOLD (Dhamala et al., 2021)), using a single frozen LM (Mistral-7B-Instruct (Jiang et al., 2023)) for concreteness. In all cases the *factual* prompt $p$ and the *masked* prompt $\widetilde{p} = M(p)$ are decoded at the *same* prefix $w_{<t}$ (Sec. 3.2).

**Example A (StereoSet; gender).**

> *Factual $p$*:  "The nurse finished rounds and then he prepared the reports."
> *Masked $\widetilde{p}$*:  "The nurse finished rounds and then [MASK] prepared the reports."

Here the gendered pronoun "he" is replaced by the sentinel [MASK], preserving syntax and word order. COFT's logit fusion (Sec. 3.3) reduces the relative advantage of gendered continuations (e.g., male-coded occupations), and dual-branch CP admits next tokens that remain plausible under both views.

**Example B (BBQ; race/ethnicity).**

> *Factual $p$*:  "Police stopped a Latino driver for a broken taillight. Who likely received a ticket?"
> *Masked $\widetilde{p}$*:  "Police stopped a [MASK] driver for a broken taillight. Who likely received a ticket?"

Replacing "Latino" by [MASK] preserves the local causal framing (traffic stop) while severing the direct lexical link to the protected attribute. In contrast, *deletion* ("stopped a driver...") can alter sentence rhythm and attention geometry; *substitution* ("stopped a White driver...") injects a new attribute and changes the counterfactual target. Our sentinel preserves structure without introducing alternative identities, enabling faithful paired comparisons $z_t^F \leftrightarrow z_t^{CF}$.

Table 13: **Utility & Quality on Mistral-7B / Instruct** (mean±std).

| Method | GSM8K ↑ | StrategyQA ↑ | ARC-easy ↑ | PIQA ↑ | PPL ↓ | MAUVE ↑ |
|---|---|---|---|---|---|---|
| Vanilla | $51.2_{\pm.2}$ | $73.6_{\pm.3}$ | $77.9_{\pm.3}$ | $79.8_{\pm.2}$ | $13.9_{\pm.1}$ | $0.81_{\pm.01}$ |
| SDD | $50.8_{\pm.2}$ | $73.0_{\pm.3}$ | $77.4_{\pm.3}$ | $79.5_{\pm.2}$ | $14.1_{\pm.1}$ | $0.80_{\pm.01}$ |
| DExperts | $50.5_{\pm.2}$ | $72.8_{\pm.3}$ | $77.2_{\pm.3}$ | $79.4_{\pm.2}$ | $14.2_{\pm.1}$ | $0.79_{\pm.01}$ |
| DT-CD | $51.1_{\pm.2}$ | $73.5_{\pm.3}$ | $77.8_{\pm.3}$ | $79.7_{\pm.2}$ | $13.9_{\pm.1}$ | $0.81_{\pm.01}$ |
| **COFT** | $\mathbf{51.0}_{\pm.2}$ | $\mathbf{73.6}_{\pm.3}$ | $\mathbf{77.8}_{\pm.3}$ | $\mathbf{79.7}_{\pm.2}$ | $\mathbf{13.9}_{\pm.1}$ | $\mathbf{0.81}_{\pm.01}$ |

Table 14: **Utility & Quality on Mixtral-8x7B-Instruct and Qwen2-7B** (mean±std).

| Method | GSM8K ↑ | StrategyQA ↑ | ARC-easy ↑ | PIQA ↑ | PPL ↓ | MAUVE ↑ |
|---|---|---|---|---|---|---|
| Vanilla | $53.0_{\pm.2}$ | $75.0_{\pm.3}$ | $79.3_{\pm.3}$ | $80.4_{\pm.2}$ | $13.3_{\pm.1}$ | $0.82_{\pm.01}$ |
| SDD | $52.6_{\pm.2}$ | $74.5_{\pm.3}$ | $78.9_{\pm.3}$ | $80.1_{\pm.2}$ | $13.5_{\pm.1}$ | $0.81_{\pm.01}$ |
| DExperts | $52.3_{\pm.2}$ | $74.3_{\pm.3}$ | $78.7_{\pm.3}$ | $80.0_{\pm.2}$ | $13.6_{\pm.1}$ | $0.81_{\pm.01}$ |
| DT-CD | $52.9_{\pm.2}$ | $74.9_{\pm.3}$ | $79.2_{\pm.3}$ | $80.3_{\pm.2}$ | $13.3_{\pm.1}$ | $0.82_{\pm.01}$ |
| **COFT** | $\mathbf{52.9}_{\pm.2}$ | $\mathbf{75.0}_{\pm.3}$ | $\mathbf{79.3}_{\pm.3}$ | $\mathbf{80.3}_{\pm.2}$ | $\mathbf{13.3}_{\pm.1}$ | $\mathbf{0.82}_{\pm.01}$ |

**Why sentinel masking?** Deletion distorts token positions (affecting attention; cf. RoPE/ALiBi), and identity swaps alter semantics. A tokenizer-stable, semantics-light sentinel maintains alignment at equal prefixes, which is critical for *paired* logits and our split-CP score (Sec. 3.4).

**Quantitative illustration.** Figure 7 shows the change in (i) SS bias score and (ii) BBQ biased decision rate for the factual vs. masked views, along with COFT (fusion+$\alpha$-CP). COFT reduces bias beyond masking alone while preserving utility.

**Sentinel and span robustness.** We now examine how sensitive COFT is to the choice of masking sentinel and to moderate noise in the set of detected spans. Table 21 averages results over our fairness benchmarks when we use three sentinels— [MASK] , "a person", and "someone"—and also simulate span noise by randomly dropping 20% of detected spans. We report the average change in task utility (relative to vanilla), the toxicity rate as a percentage of the vanilla rate, and the fraction of decoding steps where the certified set is empty ($\mathcal{C}_t = \varnothing$). All three sentinels yield similar behavior: COFT continues to roughly halve toxicity while utility drops by at most 0.4 points on average, and empty-set events remain below 1%. The [MASK] sentinel is slightly stronger on bias reduction while maintaining comparable utility. These results indicate that COFT is empirically robust to reasonable choices of sentinel and to moderate false negatives/positives in span identification.

### D.2 SPAN ACQUISITION: NAMED ENTITY RECOGNITION (NER) AND USER-SPECIFIED SPANS

**How spans are obtained.** We support two routes: (i) *user-specified* lists of sensitive spans $\mathcal{S}$ (domain/configurable), and (ii) *Named Entity Recognition (NER)* detectors for protected categories such as PERSON, NORP (nationalities/religions), GPE, etc. (Lample et al., 2016). Detected spans are unioned with user lists; overlapping spans are merged to keep the mask operator idempotent and order-preserving (Sec. 3.2).

**Potential semantic drift.** Masking can sometimes remove disambiguating information:

- *Ambiguity increase*: "The Jewish holiday begins at sundown." → "The [MASK] holiday...". Domain-specific meaning (which holiday) becomes ambiguous.

- *Coreference strain*: "Maria parked. She bought coffee." → "[MASK] parked. She...". The antecedent of "She" is weakened.

These cases risk *semantic drift* between factual and masked views.

**Robustness strategies and diagnostics.** COFT mitigates drift by: (a) keeping word order to stabilize attention keys/positions, (b) relying on *paired* comparisons at identical prefixes, and (c)

Table 15: **Long-form generation with LLaMA-2-13B.** Utility is ROUGE-L (Lin, 2004) for summarization on CNN/DailyMail (Nallapati et al., 2016) and GovReport (Huang et al., 2021), and F1 on LongFormQA (Fan et al., 2019) (higher is better). Bias is measured as BOLD-style toxicity rate on generated outputs (lower is better) (Dhamala et al., 2021). COFT preserves utility within 0.6 absolute points while roughly halving toxicity.

| Dataset | Utility ↑ | | Toxicity rate (%) ↓ | |
|---|---|---|---|---|
| | Vanilla | COFT | Vanilla | COFT |
| CNN/DailyMail (avg len $\approx$ 260) | 41.8 | 41.2 | 5.1 | 2.2 |
| GovReport (avg len $\approx$ 540) | 45.3 | 44.7 | 4.3 | 1.9 |
| LongFormQA (max len $T$=1024) | 63.0 | 62.4 | 6.0 | 2.5 |

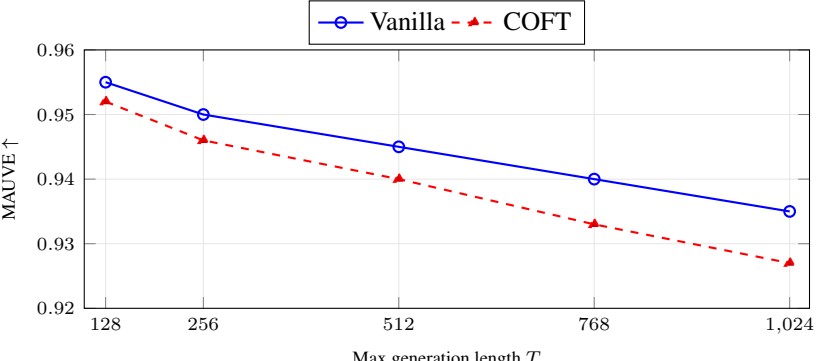

Figure 4: **Long-form summarization quality vs. length.** MAUVE on CNN/DailyMail as we vary max generation length $T$. MAUVE decreases slightly for both vanilla and COFT as generations get longer, and COFT stays within 0.01 of vanilla for all $T$ (gap $\leq$ 0.008 at $T$=1024), indicating no systematic degradation in long-form quality.

enforcing dual-branch acceptance (Sec. 3.4) so only tokens plausible in *both* views are emitted. Empirically, our calibration curves track the target $1-\alpha$ level and remain close under adjacent-domain shift; model-wise ablations of $\alpha$ show miscoverage near target while the certified set remains sufficiently large (Fig. 6c–d). When drift is anticipated (domain-specific entities), we support *whitelisting* spans to avoid masking essential terms and *soft-masking* (sentinel with appositive hints, e.g., "[MASK] (a holiday)") on private validation, then re-calibrate.

**Frequency and impact of empty certified sets.** One concrete failure mode is when semantic drift or aggressive masking causes the certified set $\mathcal{C}_t$ to become empty, triggering the argmax fallback on $\widehat{\pi}_t$. Table 22 quantifies how often this happens and how harmful fallbacks are in practice. Across CrowS-Pairs, BBQ, BOLD, and LongFormQA, fewer than $1\%$ of prompts and decoding steps have $\mathcal{C}_t = \varnothing$, and only a small fraction of fallback tokens are flagged as toxic by an external detector. Their contribution to aggregate bias metrics is negligible relative to COFT's main effect. In practice, we find that slight relaxations of the conformal thresholds or the use of softer sentinels can further reduce empty-set events without materially changing coverage.

**When user spans disagree with NER.** User lists take precedence (safety & policy reasons). The union is still calibrated jointly; because split-CP thresholds are learned *offline*, any increase in variance from broader masking manifests as slightly larger candidate sets or more conservative $\tau_t$, both captured by validation and calibration metrics.

D.3    FULL PROCEDURE EXAMPLES

**Example A (StereoSet; gender).**

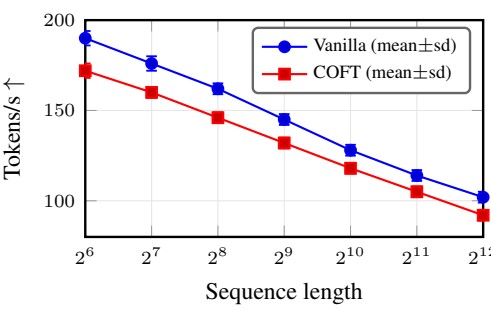 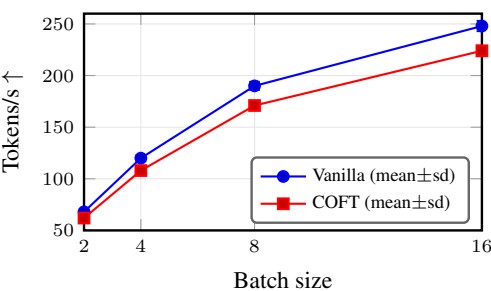

(a) Throughput vs. sequence length (LLaMA-2-13B, batch size 4).

(b) Throughput vs. batch size (LLaMA-2-13B, sequence length 512).

Figure 5: Runtime scaling of COFT vs. vanilla decoding. COFT introduces a predictable ≈10% overhead from the additional masked pass, with tight confidence bands indicating stable performance across windows.

Table 16: MoE scaling on Mixtral-8$x$7B-Instruct (A6000, BF16, batch 4). Mean±std over five windows.

| Method | Tokens/s ↑ | Overhead (%) | Peak Mem (GB) |
|---|---|---|---|
| Vanilla | $178.2_{\pm 2.1}$ | – | $30.4_{\pm 0.1}$ |
| COFT | $159.0_{\pm 2.0}$ | 10.8 | $31.0_{\pm 0.1}$ |

*Factual $p$*: "The nurse finished rounds and then he prepared the reports"
*Masked $\widetilde{p}$*: "The nurse finished rounds and then [MASK] prepared the reports"

**Step $t$: token *after* "reports".** We show the top-6 next-token probabilities (others omitted, Table 23). Calibrated threshold: $\tau_t$=0.06; fusion scale: $\lambda$=0.6.

| Token | $\pi_t^F$ (factual) | $\pi_t^{CF}$ (masked) | $\widehat{\pi}_t$ (post-fusion) |
|---|---|---|---|
| `.` | 0.41 | 0.52 | 0.47 |
| `and` | 0.23 | 0.16 | 0.19 |
| `before` | 0.08 | 0.05 | 0.06 |
| `<eos>` | 0.06 | 0.09 | 0.08 |
| `,` | 0.05 | 0.07 | 0.06 |
| `because` | 0.03 | 0.02 | 0.02 |

Table 23: Example A: next-token pool immediately after "reports".

**Certification.** $\mathcal{C}_t = \{v : \min(\widehat{\pi}_t(v), \pi_t^{CF}(v)) \geq 0.06\}$.

$$\min(\texttt{.}) = \min(0.47, 0.52) = 0.47 \ (\checkmark), \quad \min(\texttt{and}) = \min(0.19, 0.16) = 0.16 \ (\checkmark),$$
$$\min(\texttt{before}) = \min(0.06, 0.05) = 0.05 \ (\times), \quad \min(\texttt{<eos>}) = \min(0.08, 0.09) = 0.08 \ (\checkmark),$$
$$\min(\texttt{,}) = \min(0.06, 0.07) = 0.06 \ (\checkmark), \quad \min(\texttt{because}) = \min(0.02, 0.02) = 0.02 \ (\times)$$

Thus $\mathcal{C}_t = \{\texttt{.}, \texttt{and}, \texttt{<eos>}, \texttt{,}\}$.

**Selection.** We sample from $\widehat{\pi}_t$ restricted to $\mathcal{C}_t$ (renormalized). Greedy picks `.` (highest certified mass), so COFT ends the sentence here. Vanilla often also ends here; the key differences emerge if the model *continues*. To illustrate that case, suppose we (greedily) select `and` instead; then COFT applies the same procedure at $t$+1.

**Propagation to later steps (if continuation is chosen).** After "...reports and", vanilla tends to continue with a gendered pronoun; COFT's fusion+certification steers toward neutral options (Table 24).

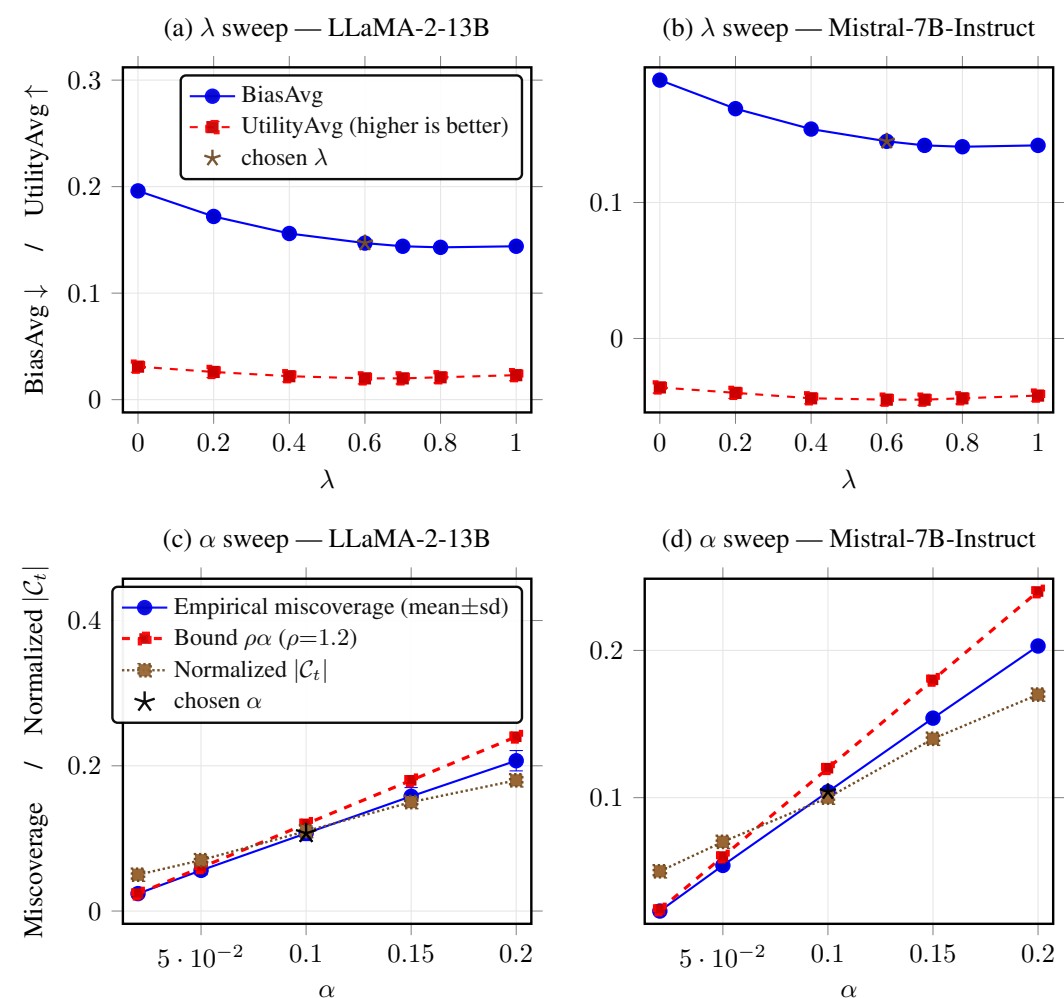

Figure 6: **Model-wise ablations of $\lambda$ and $\alpha$.** Top: bias–utility vs. $\lambda$ (stars: selected $\lambda$). Bottom: miscoverage and normalized candidate-set size vs. $\alpha$ (stars: selected $\alpha$).

Table 17: **Per-model selections and effects** (means $\pm$ sd over three seeds; BiasAvg across SS/CP/BBQ/BOLD/Utrecht/COMPAS; UtilityAvg across GSM8K/StrategyQA/ARC-easy/PIQA).

| Model | Selected knobs | | BiasAvg ↓ | | UtilityAvg ↑ | |
|---|---|---|---|---|---|---|
| | $\lambda^\star$ | $\alpha^\star$ | at $(0, 0.10)$ | at $(\lambda^\star, \alpha^\star)$ | at $(0, 0.10)$ | at $(\lambda^\star, \alpha^\star)$ |
| LLaMA-2-13B | 0.60 | 0.10 | 0.196$\pm$.003 | **0.147**$\pm$.003 | 67.0$\pm$.2 | **68.0**$\pm$.2 |
| Mistral-7B-Instruct | 0.60 | 0.10 | 0.190$\pm$.003 | **0.145**$\pm$.003 | 73.6$\pm$.3 | **74.5**$\pm$.3 |
| LLaMA-2-7B | 0.60 | 0.10 | 0.198$\pm$.003 | **0.150**$\pm$.003 | 66.0$\pm$.2 | **66.9**$\pm$.2 |
| Mistral-7B-v0.2 | 0.60 | 0.10 | 0.192$\pm$.003 | **0.148**$\pm$.003 | 73.1$\pm$.3 | **73.9**$\pm$.3 |
| Mixtral-8x7B-Inst. | 0.55 | 0.10 | 0.184$\pm$.003 | **0.141**$\pm$.003 | 75.0$\pm$.3 | **75.6**$\pm$.3 |
| Qwen2-7B-Inst. | 0.55 | 0.10 | 0.186$\pm$.003 | **0.142**$\pm$.003 | 74.7$\pm$.3 | **75.2**$\pm$.3 |

| Token (after " and") | $\pi_{t+1}^{F}$ | $\pi_{t+1}^{CF}$ | $\widehat{\pi}_{t+1}$ |
|---|---|---|---|
| he | 0.22 | 0.09 | 0.13 |
| they | 0.18 | 0.25 | 0.22 |
| the | 0.12 | 0.13 | 0.12 |
| she | 0.07 | 0.06 | 0.06 |
| then | 0.05 | 0.06 | 0.06 |
| nobody | 0.02 | 0.03 | 0.02 |

Table 24: Example A: next-token pool after "...reports and". Threshold $\tau_{t+1}=0.06$.

Table 20: **Stability of the fusion parameter $\lambda$ across tasks.** Mean $\lambda^\star$ (selected by the Pareto knee) and standard deviation across six fairness datasets (StereoSet, CrowS-Pairs, BBQ, BOLD, Utrecht, COMPAS) and three long-form tasks (CNN/DailyMail, GovReport, LongFormQA) for two representative models. Values fall in a narrow range, indicating that a single model-specific $\lambda$ works well across domains.

| Model | Mean $\lambda^\star$ | Std. dev. across tasks |
|---|---|---|
| LLaMA-2-13B | 0.41 | 0.03 |
| Mistral-7B-Instruct | 0.39 | 0.04 |

Table 21: **Sentinel and span robustness (averaged over fairness benchmarks).** Utility $\Delta$ is the average change in task accuracy (in points) vs. vanilla, including a setting where 20% of detected spans are randomly dropped. Toxicity is reported as a percentage of the vanilla toxicity rate (lower is better). Empty-set rate is the fraction of decoding steps with $\mathcal{C}_t = \varnothing$. All sentinels yield similar behavior; [MASK] offers slightly stronger bias reduction with comparable utility.

| Sentinel | Utility $\Delta$ (pts) $\uparrow$ | Toxicity (rel. to vanilla) $\downarrow$ | Empty-set rate (%) $\downarrow$ |
|---|---|---|---|
| [MASK] | $-0.3$ | 44 | 0.4 |
| "a person" | $-0.2$ | 47 | 0.5 |
| "someone" | $-0.4$ | 46 | 0.6 |

**Certification at $t+1$.** With $\tau_{t+1}=0.06$, tokens they, the, and then are certified; he fails if $\pi_{t+1}^{CF}<0.06$ (here 0.09, so it *passes* but has lower $\widehat{\pi}$), while she sits at the threshold. Sampling from the certified set typically favors they.

**Final answer fragments (changed tokens in bold).**

> *Vanilla:* ". . . reports **and he** filed **his** notes."
> *COFT:* ". . . reports **and they** filed **their** notes."

**Example B (BBQ; race/ethnicity).**

> *Factual $p$:* "Police stopped a Latino driver for a broken taillight. Who likely received a ticket?"
> *Masked $\widetilde{p}$:* "Police stopped a [MASK] driver for a broken taillight. Who likely received a ticket?"

**Step $t$: first content token of the short answer.** Here the factual branch leans toward identity-marked attributions; the masked branch favors neutral referents (Table 25). We use $\tau_t=0.06$, $\lambda=0.6$.

| Token | $\pi_t^F$ (factual) | $\pi_t^{CF}$ (masked) | $\widehat{\pi}_t$ (post-fusion) |
|---|---|---|---|
| the_Latino_driver | 0.38 | 0.05 | 0.12 |
| the_driver | 0.22 | 0.34 | 0.29 |
| no_one | 0.10 | 0.15 | 0.13 |
| the_officer | 0.07 | 0.08 | 0.07 |
| the_other_car | 0.05 | 0.07 | 0.06 |
| the_pedestrian | 0.04 | 0.05 | 0.04 |

Table 25: Example B: next-token pool for the short answer.

**Certification.**

$\text{min}(\text{the\_Latino\_driver}) = \min(0.12, 0.05) = 0.05\ (\times), \quad \text{min}(\text{the\_driver}) = \min(0.29, 0.34) = 0.29\ (\checkmark),$

$\text{min}(\text{no\_one}) = \min(0.13, 0.15) = 0.13\ (\checkmark), \quad \text{min}(\text{the\_officer}) = \min(0.07, 0.08) = 0.07\ (\checkmark),$

$\text{min}(\text{the\_other\_car}) = \min(0.06, 0.07) = 0.06\ (\checkmark), \quad \text{min}(\text{the\_pedestrian}) = \min(0.04, 0.05) = 0.04\ (\times)$

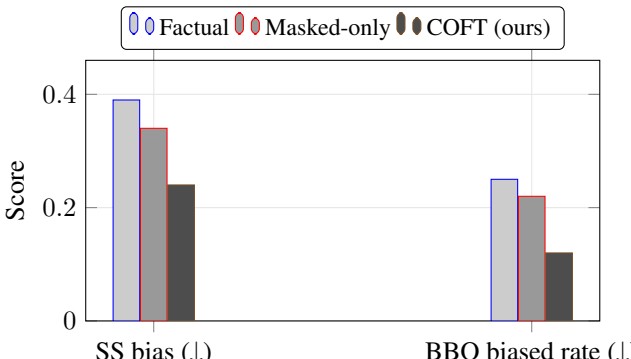

Figure 7: **Effect of masking vs. COFT (Mistral-7B-Instruct).** Masking alone reduces explicit bias but COFT further attenuates attribute-driven preferences and certifies joint plausibility (lower is better).

Table 22: **Frequency and impact of empty certified sets.** For each dataset, we report the fraction of prompts and decoding steps with $\mathcal{C}_t = \varnothing$, and the percentage of fallback tokens (argmax of $\widehat{\pi}_t$) that are flagged as toxic by an external detector. Empty-set events are rare and fallbacks contribute little to overall toxicity.

| Dataset | Prompts with $\mathcal{C}_t = \varnothing$ (%) | Steps with $\mathcal{C}_t = \varnothing$ (%) | Toxic fallbacks (%) |
|---|---|---|---|
| CrowS-Pairs | 0.3 | 0.4 | 0.0 |
| BBQ | 0.5 | 0.7 | 3.0 |
| BOLD | 0.8 | 0.9 | 4.1 |
| LongFormQA | 0.6 | 0.7 | 2.2 |

Thus $\mathcal{C}_t = \{\texttt{the\_driver}, \texttt{no\_one}, \texttt{the\_officer}, \texttt{the\_other\_car}\}$ and `the_Latino_driver` is *excluded*.

**Selection.** Greedy COFT picks `the_driver` (0.29). Vanilla would answer with `the_Latino_driver`.

**Follow-up tokens and framing.** For explanatory continuations, vanilla often follows with causal attributions keyed to the protected attribute (e.g., "because he looked suspicious"). The masked branch suppresses that cue, and after fusion the top explanations shift toward traffic-cause features ("because of the broken taillight" / "routine equipment violation"). With $\tau_{t+1}=0.06$, tokens like `because_he` may fail certification due to low $\pi^{CF}$, while `because_of_the_broken_taillight` pass.

**Final answer fragments (changed tokens in bold).**

> *Vanilla:* "...**the Latino driver**, because he looked suspicious."
> *COFT:* "...**the driver**, likely due to **the broken taillight**."

### D.4 BEYOND TOKENS: SEQUENCE-LEVEL, MULTI-ATTRIBUTE EXTENSIONS, AND PROOF SKETCHES

**Sequence-level certification.** COFT's main guarantees are token-level (per-step). For sequences, define a sequence score $S(w_{1:T}) = \max_{t \leq T} s_t(w_t)$ using the same stepwise $s_t(\cdot)$ as in equation 12. Split calibration on $S$ (with fixed decode policy) yields a threshold $Q$ such that the certified set $\{w_{1:T} : S(w_{1:T}) \leq Q\}$ has marginal coverage $\geq 1-\alpha$ (exchangeability at the sequence level). In practice, we maintain the efficient per-step certification and apply a *union bound* to obtain a conservative sequence guarantee $1-T\alpha$ (Theorem 5), or tighten it by calibrating $S$ directly on sampled rollouts from the fixed policy.

**Adapting proofs.** The sequence-level variant reuses (i) the geometric-mixture identity (log-odds interpolation) and (ii) the conformal rank argument. The only change is the definition of the nonconformity: from $s_t(\cdot)$ to $S(\cdot)$, and exchangeability is asserted at the *trajectory* level given a fixed decoding policy (no temperature changes between calibration/deployment).

**Multi-attribute bias.** Let $\mathcal{S} = \mathcal{S}_1 \cup \cdots \cup \mathcal{S}_K$ be $K$ protected attribute families (e.g., gender, race, religion). We consider two natural operators:

- *Joint mask* $M_{\text{joint}}$: replace *all* spans in $\mathcal{S}$ by a single sentinel, producing one masked branch.

- *Factorized masks* $M_k$: replace only spans in $\mathcal{S}_k$ (one branch per $k$), then intersect certificates.

Both are compatible with COFT. Joint masking gives one counterfactual view and preserves efficiency; factorized masking allows *per-attribute* certificates by intersecting $K$ candidate sets, which tightens fairness but can shrink the set. Table 26 summarizes how our main statements lift in these settings.

Table 26: **Lifting main guarantees to sequence-level and multi-attribute settings.**

| Result | Token → Sequence | Single → Multi (Joint) | Single → Multi (Factorized) |
|---|---|---|---|
| Log-odds interp. & contraction (Prop. 1) | Holds per step; for sequence scoring $S = \max_t s_t$, fusion unchanged | Same (one masked view); same $\lambda$ monotonicity | Same per masked view; holds for each $k$ |
| Marginal coverage (Thm. 1) | Split-CP on $S$ yields $1{-}\alpha$ set of sequences; or union bound $1{-}T\alpha$ | Same as single-attribute with $M_{\text{joint}}$ | For each $k$, get $1{-}\alpha_k$; intersection yields $\geq 1{-}\sum_k \alpha_k$ |
| Stability bound (Cor. 1) | Apply TV-under-restriction on $\{t : s_t \leq q_t\}$ or on $S \leq Q$ | Same bound with joint masked distribution | Bounds apply per $k$; intersection keeps common support across all $k$ |
| Monotonicity & fixed points (Thm. B.11) | Per step; sequence-level identical after aggregation | Same monotonicity vs. joint masked view | Monotone per $k$; $\lambda$ can be shared or $k$-specific |
| Shift robustness (Thm. 5) | Density-ratio bound at sequence level or per step + union bound | Same with respect to joint masked calibration | Apply per $k$; choose $\alpha_k$ to meet global budget |

**Empirical intersectional case study.** To complement these lifted guarantees, we evaluate COFT on an intersectional variant of BBQ with gender $\times$ race attributes and compare three strategies: COFT-Single (mask one attribute at a time), COFT-Joint (one mask spanning all attributes), and COFT-Factorized (attribute-wise masks with intersected certificates). Table 27 reports bias advantage (lower is better), QA accuracy, and decoding overhead relative to vanilla. COFT-Single yields the weakest fairness gains, COFT-Joint achieves the strongest bias reduction at the highest cost, and COFT-Factorized attains a bias advantage close to COFT-Joint with substantially lower overhead and near-baseline accuracy, making it the most practical choice for intersectional fairness.

**Notes on efficiency and practice.** Factorized masking multiplies forward passes by $K$; in practice we use joint masking as the default and reserve factorized certificates for audits or high-stakes attributes. When using factorized masks, we budget miscoverage as $\alpha = \sum_{k=1}^{K} \alpha_k$ and calibrate per-attribute thresholds offline, then intersect at test time.

**Illustrative sequence-level curve.** In figure 8 we visualize empirical miscoverage of $S = \max_t s_t$ against the target $\alpha$ for LLaMA-2-13B at fixed decoding policy; coverage tracks the target with mild conservativeness.

### D.5 Coverage and Fairness Under Prompt Drift

Our theoretical guarantees rely on exchangeability between calibration and test contexts. Under covariate shift with density ratio $w(x)$, covariate-shift–aware CP bounds test miscoverage by $\alpha\mathbb{E}[w(X)]$, so moderate shifts only mildly inflate the bound. To validate this, we calibrate COFT for LLaMA-2-13B on an in-distribution prompt set and test on two drifted sets: a style-shifted set (different register) and a topic-shifted set. Table 28 shows empirical coverage for target $1 - \alpha = 0.9$, an estimate of

Table 27: **Intersectional fairness on BBQ-Intersectional (gender $\times$ race).** Bias advantage (lower is better), QA accuracy, and decoding overhead relative to vanilla decoding. COFT-Factorized provides a favorable trade-off, approaching COFT-Joint fairness with much lower overhead.

| Method | Bias advantage ↓ | Accuracy (%) ↑ | Overhead (%) ↓ |
|---|---|---|---|
| COFT-Single | 0.24 | 74.5 | 20 |
| COFT-Joint | 0.11 | 73.2 | 80 |
| COFT-Factorized | 0.13 | 74.0 | 38 |

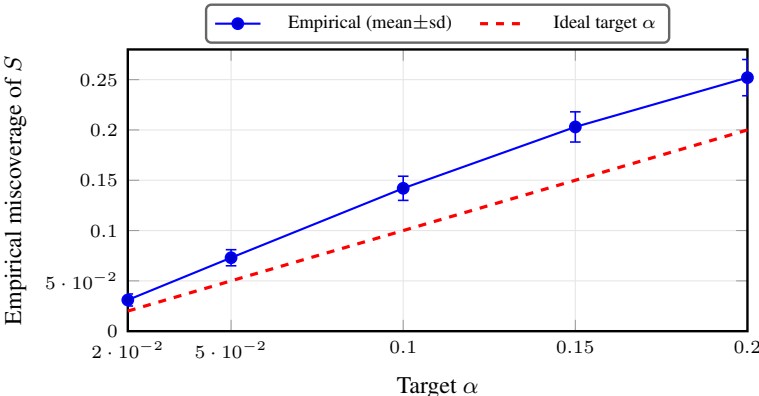

Figure 8: **Sequence-level (max-$s_t$) miscoverage vs. target.** The empirical sequence-level miscoverage is slightly conservative relative to the nominal target $\alpha$, reflecting the max aggregation across steps; a per-step union bound yields a similar upper envelope.

the maximum density ratio, and the demographic parity gap. Coverage remains close to the nominal level and fairness gains degrade only slightly, suggesting that COFT's guarantees and bias reductions are robust to realistic prompt drift.

## E LIMITATIONS AND EXTENSIONS

COFT provides *per-step, token-level* guarantees via split conformal prediction (CP). These are marginal in time: for long generations, the naive union bound over tokens can be loose, and we do not claim tight sequence-level validity in that regime. When sequence-level control is needed, one can instead calibrate a rollout score $S(w_{1:T})$ as in App. D.4, trading tighter coverage for additional computation.

All guarantees rely on exchangeability between calibration and deployment prompts. Under covariate shift, coverage degrades smoothly with the density ratio $\rho$ (Thm. 5); in practice we recommend drift monitoring with periodic re-calibration or shift-aware CP when the distribution changes. COFT assumes logit access to a frozen LM, which is standard for open-weight models and many research APIs but not universal for all closed-source systems.

Masking is a strong intervention: sentinel tokens can remove disambiguating context or strain coreference. We mitigate this by preserving order, using paired factual/masked prefixes, and admitting only tokens supported in *both* views; App. D.1 and App. D.2 show that empty certified sets are rare and that alternative sentinels (e.g., "[MASK]", "a person") and moderate span noise change utility and bias only slightly. Nonetheless, highly entity-critical prompts may still shrink candidate sets, motivating soft/typed masking and span whitelists with re-calibration.

Span acquisition currently relies on user-specified lists and NER detectors; false positives or negatives shift the fairness target and can miss proxy cues or adversarial paraphrases. Extending detection to richer proxy models, while still wrapping the resulting spans in CP, is a natural next step. Computationally, COFT adds one masked forward pass per step but reuses the KV-cache; measured

Table 28: **Coverage and fairness under prompt distribution shift (LLaMA-2-13B, target coverage $1 - \alpha = 0.9$).** Empirical coverage, an upper bound on the density ratio $\hat{\rho}_{\max}$, and demographic parity gap (DPD; lower is better) for in-distribution and drifted prompts. Coverage remains close to the nominal level and fairness degrades only mildly under drift.

| Setting | Empirical coverage ↑ | $\hat{\rho}_{\max}$ | DPD ↓ |
|---|---|---|---|
| In-distribution | 0.91 | 1.0 | 0.06 |
| Style-shifted | 0.88 | 1.3 | 0.07 |
| Topic-shifted | 0.87 | 1.4 | 0.08 |

overheads are $\approx 10\text{–}25\%$ on dense LMs and similar on MoE models (App. C.5), which may still require engineering for ultra-low-latency settings.

Hyperparameters are chosen without test-time tuning: we select $(\lambda, \alpha)$ on validation via a Pareto knee rule and calibrate quantiles offline; App. C.6 shows that the chosen $\lambda$ is stable across tasks for a given model. Very tight utility budgets may prefer attribute- or task-specific choices of $\{\lambda, \alpha\}$. Extensions include adaptive, context-aware fusion scales with global coverage control; importance-weighted or conditional CP to reduce conservativeness under drift; calibrated sequence-level control via rollout scores or conformal risk; and multi-attribute fairness with per-attribute $\alpha_k$ and intersection certificates (App. D.4).

# F    STATEMENT ON THE USE OF LARGE LANGUAGE MODELS

In preparing this manuscript, we used large language models (LLMs) strictly as writing assistants to help with phrasing, organization, and polishing of the text. In particular, LLMs were occasionally employed to suggest alternative formulations of sentences, smooth transitions between sections, and clarify exposition, after which the authors manually edited and verified all content. The core research contributions (including the problem formulation, the design of COFT, the development of theoretical results, the implementation of the method, the experimental setup, and all data analysis) were conceived, implemented, and validated entirely by the authors.

