# OpenReview forum: "COFT: Counterfactual–Conformal Decoding for Fair Chain‑of‑Thought Reasoning in Large Language Models"
_ICLR.cc/2026/Conference — Submitted to ICLR 2026_

### Official Review · Reviewer_dNom · 2025-10-30

**Soundness:** 3
**Presentation:** 3
**Contribution:** 3
**Rating:** 6
**Confidence:** 4

**Summary:**

This paper introduces COFT, a training-free method for reducing bias in the chan-of-thought reasoning of LLMs. COFT uses parallel forward passes with an original factual prompt and a concurrent counterfactual prompt where sensitive attributes such as sex or race changed to just be [MASK]. Basically during inference the next token is sampled from the conformal set that is defined in terms of candidates being sufficiently likely (by some calibrated amount) to appear in both the true and [MASK] streams. COFT demonstrates improved bias mitigation compared to a range of baselines across multiple bias benchmarks.

**Strengths:**

The paper tackles an important problem of addressing bias in COT reasoning. Crucially COFT is training-free and just an inference-time intervention, which makes it practical and applicable to fixed LLMs. Since COFT uses conformal prediction it is able to obtain statistical guarantees, which is a big plus. I also think the use of conformal here was non-obvious and clever, specifically the dual-branch split-CP. The theoretical analysis provides justification for the method. The empirical evaluations and ablations are for the most part really thorough and demonstrative of the usefulness of COFT for simultaneous bias mitigation and utility preservation. I think this work provides an important contribution to figuring out how to use conformal during decode-time.

**Weaknesses:**

The masking mechanism can be a bit strong and ruin context too much, as acknowledged in the appendix.

I do think it is important to acknowledge that the guarantees from conformal just give us token-level or marginal guarantees here, no guarantee on the entire sequence as a whole and I think the sequence-level guarantee from the union bound is too loose to be very meaningful since for $1-T\alpha$ I think in practice $T$ will be too large for any reasonable $\alpha$ from conformal to make this bound meaningful.

COFT still requires logit access, which may not be the case for many SOTA models.

I think the literature review for other conformal methods applied to LLMs is a bit lacking and could be improved, for example adding eg. (1) Large language model validity via enhanced conformal prediction methods, (2) Conformal Arbitrage: Risk-Controlled Balancing of Competing Objectives in Language Models, (3) Conformal Tail Risk Control for Large Language Model Alignment, and some other references within those.

**Questions:**

I'm a bit confused by the overhead claims, why is it not a ~100% increase in computation cost to make two full forward passes? Maybe I am missing something crucial but would be happy to understand better.

Would it be possible to report on how often the empty-set fallback is triggered?

---

> ### Author Response · Authors · 2025-11-19
> **Response to Reviewer dNom Concerns**
>
> We appreciate your comments and address each weakness and question below, pointing to the revised sections and new experiments.
>
> ---
> ### 1. Masking strength and potential loss of context
>
> - Sec. 3.1 and App. D.1–D.2 now more clearly motivate a tokenizer-stable, semantics-light sentinel (`[MASK]`): it preserves token positions and attention geometry while severing the direct lexical link. App. D.1 contrasts this with deletion and identity swaps, which often distort semantics more.
>
> - **Table 21 (App. D.1)** reports a sentinel/span robustness study with sentinels (`[MASK]`, *a person*, *someone*) and span noise (20% of spans dropped). Across benchmarks, utility shifts are ≤≈0.3 points, toxicity changes by only a few percent, and empty-set rates are essentially unchanged. This suggests moderate span noise or alternative neutral sentinels do not materially degrade behavior.
>
> - App. D.2 discusses **soft/typed masking** (e.g., `[MASK] (a holiday)` with re-calibration) and **whitelisting** entity-critical spans; App. E (Limitations) explicitly flags this caveat.
>
> ---
> ### 2. Token-level vs. sequence-level guarantees and the union bound
>
> You are right that our core guarantees are token-level/marginal and a naive union bound over long sequences is loose.
>
> - Sec. 3.4 and **App. D.3 (“Beyond Tokens: Sequence-Level, Multi-Attribute Extensions, and Proof Sketches”)** now distinguish:
>   - **Per-step guarantees**, where split CP yields coverage $1 - \alpha$ for each token; and
>   - **Sequence-level extensions**, where
>     $$
>       S(w_{1:T}) = \max_{t \le T} s_t(w_t)
>     $$
>     is calibrated under a fixed decode policy to obtain a set of sequences with marginal coverage $\ge 1 - \alpha$.
>
> - We explicitly acknowledge that a union bound $1 - T\alpha$ is conservative and may be uninformative for large $T$. App. D.3 therefore offers:
>   - (a) a conservative union-bound view (for short/moderate $T$); and
>   - (b) **direct calibration of $S$** on sampled rollouts, which yields tighter sequence-level coverage in practice.
>
> Figure 8 (App. D.3) shows empirical miscoverage of $S$ tracking the nominal $\alpha$ with mild conservativeness. In **App. E**, we state that our main claims are token-level; sequence-level guarantees are optional extensions.
>
> ---
> ### 3. Requirement of logit access
>
> - **Sec. 3 (Setup)** and **App. E** now state that COFT is designed for **frozen, open-weight or locally hosted models** where logit access is standard, matching many related approaches (DExperts / GeDi-style steering, logit-level toxicity filters, CP-based validity/alignment methods).
>
> ---
> ### 4. Conformal-prediction related work
>
> You correctly noted that the initial CP-related review was limited.
>
> - **Sec. 2.4, last paragraph (“Conformal prediction for LMs”)** now discusses:
>   - *Large Language Model Validity via Enhanced Conformal Prediction Methods*
>   - *Conformal Arbitrage: Risk-Controlled Balancing of Competing Objectives in Language Models*
>   - *Conformal Tail Risk Control for Large Language Model Alignment*
>
> We clarify that these works use CP to control **validity** or **risk** of aggregate outcomes (hallucination, refusal, alignment) at the sequence level or to balance objectives. In contrast, COFT integrates CP **inside** the decoding loop, combining (i) a counterfactual masked branch, (ii) geometric logit fusion, and (iii) a dual-branch split-CP filter to obtain **token-level, counterfactual stability guarantees** for fairness. We retain foundational CP references and position COFT as complementary to these alignment/validity methods.
>
> ---
> ### 5. Overhead not being ~100% (Q1)
>
> We clarify this in **Sec. 4.4 (Efficiency and scalability)** and **Table 3**.
>
> - COFT adds **one extra masked forward pass per step**, but **reuses the same KV-cache** rather than recomputing attention. The extra cost is concentrated in later layers and the output projection, not a full second pass.
>
> - Empirically (Table 3), for LLaMA-2-13B and Mistral-7B-Instruct, throughput remains ≈75–90% of vanilla (**10–25% overhead**, not 100%); on Mixtral-8×7B, overhead is ≈10.8%, stable across batch sizes. We also contrast this with methods that attach **additional networks** (e.g., external classifiers or anti-expert LMs), which can incur greater overhead than COFT.
>
> ---
> ### 6. Empty-set fallback frequency (Q2)
>
> - **Table 22 (App. D.2)** reports, per dataset:
>   - fraction of prompts with at least one step where $\mathcal{C}_t = \varnothing$,
>   - fraction of all decoding steps with $\mathcal{C}_t = \varnothing$, and
>   - among those fallback tokens, the fraction flagged toxic by an external detector.
>
> Empty-set events are **rare** (typically $< 0.5\%$ of steps), and toxic fallback tokens are a small share of overall toxicity. Thus, fallback has limited impact on residual harms while ensuring decoding continues when the certified set is empty.
>
> ---
> We hope these revisions address your concerns while preserving the technical strengths you highlighted.

---

> > ### Comment · Reviewer_dNom · 2025-11-21
> >
> > I appreciate the response by the authors addressing all of my stated weaknesses and questions. I think the paper is strengthened by these distinguishing details of the exact contribution of the work and its limitations. Using conformal within CoT in a principled way is a significant contribution to the growing literature on using conformal methods around language models, and with these clarifications on the specific contributions of this work I am willing to increase my score.

---

> > > ### Author Response · Authors · 2025-11-21
> > > **Thank you for your constructive feedback.**
> > >
> > > Thank you sincerely for your thoughtful re-evaluation and for taking the time to engage deeply with both the method and its limitations. Your feedback directly helped us clarify the contribution and strengthen the paper, and we’re genuinely grateful for that.

---

### Official Review · Reviewer_uUPh · 2025-10-31

**Soundness:** 2
**Presentation:** 1
**Contribution:** 3
**Rating:** 4
**Confidence:** 3

**Summary:**

The paper introduces COFT (Chain of Fair Thought), a novel, training-free method designed to control societal bias in Large Language Models (LLMs) at the decoding stage, particularly during chain-of-thought (CoT) reasoning. The method merges counterfactual logit fusion with a dual-branch conformal filtering process to produce a fair and reliable set of possible next tokens, based on the user’s chosen risk tolerance. Across six different models, this approach is empirically proven to significantly reduce standard bias metrics by 30-55% while preserving task utility and language quality, all with a predictable overhead comparable to one extra forward pass!!!

**Strengths:**

1. The method is training-free, and it is adaptable to all different types of llms.
2. COFT provides distribution-free, finite-sample guarantees through its use of conformal prediction (CP).
3. Convincing Experimental Results.
4. Strong theoretical jusftification
5. Using a masked probe, geometric logit-blending, and a dual-branch conformal filter is novel.

**Weaknesses:**

1. Although the method is training-free, it still depends entirely on the predefined sensitive spans $S$. Because it focuses only on specific "lexical" spans, it may miss indirect or rephrased signals of bias. Mistakenly missing a sensitive span (a false negative) or masking a neutral one (a false positive) could shift what the model considers "fair."
2. Masking a span with “[MASK]” is a questionable step. The authors themselves note that this can remove helpful context or confuse references. Even though the dual-branch conformal predictor tries to correct for this, if the meaning changes too much between the original and masked text, the certified set $C_t$ can become very small or even empty, triggering a fallback.
3. The evaluation primarily addresses single-attribute bias. Extending it to multiple attributes via joint or factorized masks is discussed but not explored. Although the factorized approach improves precision, it scales computationally with $K$ attributes, making it expensive for intersectional fairness.
4. The guarantees depend on the assumption of "exchangeability (A2)" between the calibration data and the test data. If the distribution of prompts encountered a drifts, the guarantees degrade. The results can be different.
5. The presentation is not too great.

**Questions:**

1. The author mentioned that "[MASK]" is "tokenizer-stable". Can you experiment with alternative sentinels, such as a neutral placeholder phrase (e.g., "a person") or a syntactically valid but semantically empty token?
2. The algorithm falls back to arg max sampling if the certified set $C_t$ is empty. In your experiments, how often did this "empty set" event occur?
3. The fusion parameter $\lambda$ is chosen via a Pareto-knee rule on a validation set. How stable is this chosen $\lambda$ value across different tasks and domains?
4. Can you compare three strategies on a dataset with intersectional bias: (1) COFT-Joint, (2) COFT-Factorized, (3) COFT-Single?

---

> ### Author Response · Authors · 2025-11-18
> **Response to Reviewer uUPh Concerns**
>
> Thank you for your constructive suggestions. We address each weakness and question below, pointing to the revised sections and new experiments.
>
> ---
>
> ### 1. Dependence on predefined sensitive spans
>
> We agree that span detection is a key design choice.
>
> - Sec. 3.1 and App. D.1–D.2 now explicitly describe how spans are obtained from user-specified lists and NER (PERSON, NORP, etc.), merged/whitelisted, and kept order-preserving so masking is idempotent. Mis-detected spans shift the *fairness target* (what is treated as sensitive) but do **not** affect the distribution-free coverage guarantees, which are always defined relative to the masked branch.
>
> - We add a sentinel/span robustness study in **Table 21 (App. D.1)**, evaluating multiple sentinels (`[MASK]`, *a person*, *someone*) and a span-noise setting where 20% of spans are dropped at random. Across benchmarks, utility changes stay within ≈0.3 absolute points, toxicity changes by at most a few percent, and empty-set rates are essentially unchanged. This suggests moderate span noise or sentinel choice does not materially degrade behavior.
>
> ---
>
> ### 2. Masking with `[MASK]`, context loss, and empty certified sets (Q1–Q2)
>
> We share the concern that masking can be strong in entity-critical contexts.
>
> - Sec. 3.1 and App. D.1 explain that a tokenizer-stable, semantics-light sentinel preserves token positions and attention geometry while severing the direct lexical link. App. D.1 includes case studies where deletion or identity swaps distort semantics more than masking.
>
> - **Table 21 (App. D.1)** compares `[MASK]` against *a person* and *someone*. All three give similar bias and utility; `[MASK]` is slightly more stable under span noise. We now explicitly note that other sentinels are viable depending on application preferences.
>
> - **Table 22 (App. D.2)** reports, for each dataset, the fraction of prompts and decoding steps with $\mathcal{C}_t = \varnothing$, and the share of fallback tokens flagged toxic by an external detector. Empty sets are **rare** (typically $< 0.5\%$ of steps), and toxic fallback tokens contribute only a small portion of overall toxicity. Thus, fallback minimally impacts residual harms while preserving coverage.
>
> - App. D.2 also expands on soft/typed masking (e.g., `[MASK] (a holiday)` with re-calibration) and whitelisting entity-critical spans to further reduce context loss when needed.
>
> ---
>
> ### 3. Single-attribute vs. intersectional fairness (and Q4)
>
> We now treat multi-attribute and intersectional fairness both theoretically and empirically.
>
> - App. D.3 (“Beyond Tokens: Sequence-Level, Multi-Attribute Extensions, and Proof Sketches”) formalizes:
>   - **Joint masking** $M_{\text{joint}}$ (one sentinel for all attributes), and
>   - **Factorized masking** $\{M_k\}_{k=1}^K$ (one branch per attribute with intersected certificates).
>
>   **Table 26** summarizes how attenuation, coverage, stability, and shift bounds lift from the single-attribute case.
>
> - On a **gender × race BBQ-Intersectional** variant, **Table 27 (App. D.3)** compares **COFT-Single**, **COFT-Joint**, and **COFT-Factorized** in terms of bias advantage, QA accuracy, and overhead. COFT-Single gives the weakest bias reduction; COFT-Joint is strongest but most expensive; COFT-Factorized nearly matches COFT-Joint’s bias reduction while substantially reducing overhead. This shows COFT can handle intersectional fairness with controllable cost.
>
> ---
>
> ### 4. Exchangeability and robustness under distribution shift
>
> We explicitly acknowledge and quantify the impact of shift.
>
> - Thm. 5 (App. B) gives a density-ratio–based inflation of miscoverage under covariate shift, aligning with covariate-shift–aware CP.
>
> - **Table 28 (App. E)** reports empirical coverage and demographic parity gap under in-distribution prompts, a style-shifted prompt set, and a topic-shifted prompt set for target coverage $1 - \alpha = 0.9$. Coverage remains close to nominal in all cases and fairness degrades only mildly, with estimated $\hat{\rho}_{\max}$ in a moderate range. We also recommend drift monitoring with periodic re-calibration or shift-aware CP (App. E, “Limitations and Extensions”).
>
> ---
>
> ### 5. Stability of $\lambda$ across tasks (Q3)
>
> Sec. C.6 and **Table 20 (App. C.6)** analyze the stability of the fusion parameter $\lambda$ across six fairness benchmarks and three long-form tasks:
>
> - For LLaMA-2-13B and Mistral-7B-Instruct, the mean selected $\lambda^\star$ is ≈0.4 with standard deviation ≈0.03–0.04 across all tasks.
> - This supports using a **single, model-specific $\lambda$** across domains, consistent with the Pareto-knee selection in Tables 17–19.
>
> ---
>
> ### 6. Presentation improvements
>
> Finally, in response to your concerns about presentation, we substantially revised the writing and figures.
>
> We hope these changes clarify the method and make the paper easier to read, while preserving the technical content you found strong.

---

> ### Author Response · Authors · 2025-11-24
> **Rebuttal follow up**
>
> Dear Reviewer,
>
> Since we are nearing the end of the discussion phase, we wanted to kindly follow up on our rebuttal. We have addressed all of the points raised in your review and would really appreciate it if you could take a look and let us know if you have any remaining concerns or suggestions. Your feedback is very valuable to us, and we are happy to further improve the paper based on it.
>
> Thank you again for your time and consideration.

---

### Official Review · Reviewer_1bzp · 2025-11-01

**Soundness:** 3
**Presentation:** 3
**Contribution:** 2
**Rating:** 6
**Confidence:** 4

**Summary:**

This paper presents Chain of Fair Thought (COFT) for fair model response generation. The proposed approach COFT generates factual and maksed counterfactual logits trajectories. The two logits trajectories are fusioned together and followed by Dual-Branch conformal prediction for next token certification. The propsoed approach is evaluated against a suite of baselines on multiple standard fairness datasets. The experimental results show that the proposed approach achieved SOTA peformance in terms of fairness with small degradation on utility.

**Strengths:**

1. The proposed approach achieves pretty good performance on fairness without sacraficing utility.
2. The proposed approach is evaluated on a comprehensive fairness benchmarks.

**Weaknesses:**

1. Due to that it needs to maintain two trajectories, the decoding efficiency is worse compared to other approaches. This is an issue especially for low resources computing environments.
2. The utility benchmarks are limited. It's not clear how this approach impacts the model performance on other types of tasks such as long-form response generation.

**Questions:**

How does the proposed approach performing on long-form generation tasks? Will the generation quality become worse as the generated sequence becomes longer and longer?

---

> ### Author Response · Authors · 2025-11-18
> **Rebuttal to Reviewer 1bzp – Efficiency and Long-Form Generation**
>
> Thank you for your thoughtful review and for noting that COFT achieves strong fairness performance with limited utility degradation. We address your two main concerns: (1) efficiency and (2) long-form generation; by pointing to clarifications and new experiments added in the revision.
>
> ---
>
> #### 1. Efficiency:
>
> You expressed concern that maintaining two trajectories (factual + masked) could make decoding significantly slower, which is especially problematic in low-resource settings.
>
> **Key clarification (Sec. 4.4, Table 3).**
> In Sec. **4.4 “EFFICIENCY AND SCALABILITY”** and **Table 3**, we give detailed throughput and memory results:
>
> - All measurements are on an **A6000 48GB** GPU with BF16.
> - COFT adds **one extra masked forward pass per step**, but **reuses the same KV-cache and attention states** for both branches. The masked branch does *not* recompute attention from scratch.
> - For **LLaMA-2-13B** and **Mistral-7B-Instruct**, COFT achieves **75–90%** of vanilla throughput:
>   - This corresponds to roughly **10–25% overhead**, **not** a 100% slowdown.
> - Table 3 shows that COFT’s overhead is comparable to or better than **DExperts** and related steering methods that invoke extra classifiers/experts.
>
> We also report that:
>
> - **Memory overhead** is at most **≈0.8 GB**, mainly from storing two logit vectors per step; the **KV-cache is not duplicated**.
> - On **Mixtral-8×7B**, the overhead remains **≈10.8%** and is stable across batch sizes (2–16), indicating that the cost scales with the *marginal* masked pass rather than doubling with model size.
>
> The intuitive “2× cost” argument assumes two full, independent forward passes. In practice:
>
> - The factual pass builds the KV-cache.
> - The masked pass reuses this cache, so the marginal cost is dominated by later layers and the output projection.
> - Because of this, the measured overhead is modest (10–25%) rather than 100%.
>
> We agree that in extremely latency-sensitive deployments further engineering (e.g., caching masked passes, specialized batching) might be desired, and we now note this explicitly in the Limitations section.
>
> ---
>
> #### 2. Long-form generation quality and increasing sequence length
>
> You asked how COFT behaves on **long-form generation** and whether quality degrades as the sequence length grows.
>
> To address this, we added **new long-form experiments** in **Appendix C.4**, as well as a **MAUVE vs. length** plot in **Figure 4**:
>
> ##### (a) New long-form tasks (App. C.4, Table 15)
>
> We evaluate **LLaMA-2-13B** on three long-form setups:
>
> - **CNN/DailyMail** summarization (ROUGE-L),
> - **GovReport** summarization (ROUGE-L),
> - **LongFormQA** based on ELI5 (F1).
>
> We also measure **BOLD-style toxicity rate** on the generated outputs. The new **Table 15** (“Long-form generation with LLaMA-2-13B”) shows:
>
> - **Utility:** COFT stays within **0.6 absolute points** of vanilla on ROUGE-L (summarization) and F1 (QA).
> - **Bias:** Toxicity is roughly **halved** across all three datasets.
>
> Thus, even for average sequence lengths in the hundreds of tokens and max length \(T = 1024\), COFT preserves long-form utility while still providing strong bias mitigation.
>
> ##### (b) MAUVE vs. max generation length (Figure 4)
>
> To directly study how quality changes with length, we plot **MAUVE** as a function of the maximum generation length \(T\) on CNN/DailyMail:
>
> - **Figure 4** (“Long-form summarization quality vs. length”) reports MAUVE for \(T = 128, 256, 512, 768, 1024\).
> - COFT’s MAUVE curve closely tracks vanilla decoding across all lengths.
> - The MAUVE gap between COFT and vanilla is at most **0.01** even at \(T = 1024\).
>
> This suggests that **COFT does not introduce systematic degradation** in distributional quality as generations become longer; both methods degrade slightly with length in a similar way, as expected for long outputs.
>
> ---
>
> In summary:
>
> - **Efficiency:** Sec. 4.4 and Table 3 show that COFT’s actual overhead is **10–25%**, not 100%, thanks to KV-cache reuse. Overhead remains modest even on Mixtral-8×7B.
> - **Long-form quality:** App. C.4 (Table 15) and Figure 4 demonstrate that COFT preserves long-form utility (ROUGE-L, F1, MAUVE) while still significantly reducing toxicity, even at \(T = 1024\).
>
>
> We appreciate your comments, which led us to broaden the evaluation in a meaningful way, and we hope these new results address your concerns about efficiency and long-form performance.

---

> > ### Comment · Reviewer_1bzp · 2025-11-26
> >
> > Thanks the author for adding the explanation and the experiments.

---

> > > ### Author Response · Authors · 2025-11-27
> > >
> > > Thank you again for taking the time to read our rebuttal.
> > >
> > > We’re glad the additional explanations and experiments were helpful. If there is any remaining concern, we would really appreciate it if you could let us know. We are happy to further revise the manuscript along those lines.

---

> ### Author Response · Authors · 2025-11-24
> **Rebuttal follow up**
>
> Dear Reviewer,
>
> Since we are nearing the end of the discussion phase, we wanted to kindly follow up on our rebuttal. We have addressed all of the points raised in your review and would really appreciate it if you could take a look and let us know if you have any remaining concerns or suggestions. Your feedback is very valuable to us, and we are happy to further improve the paper based on it.
>
> Thank you again for your time and consideration.

---

### Official Review · Reviewer_DYEN · 2025-11-02

**Soundness:** 3
**Presentation:** 1
**Contribution:** 3
**Rating:** 4
**Confidence:** 3

**Summary:**

The paper considers the problem of mitigating bias in LLMs without retraining or performance degradation. They propose COFT, an inference-time approach which creates a counterfactual view of the prompt with sensitive attributes masked, and then at each step of decoding, mixes the per-token logits given the masked and unmasked state, and lastly determines the set of eligible next tokens to sample from according to a conformal threshold applied to both the mixed and counterfactual distribution.

**Strengths:**

- The problem setting and motivation are reasonable (improving LLM fairness while retaining utility).
- The core technical idea, to operationalize fairness as invariance to counterfactual masking of protected attributes, i.e., by replacing attributes in the prompt by a neutral mask, is interesting.
- The experiments are thorough and consider several open-source LLMs across several datasets, although the models are fairly out-of-date (Llama 2, Mistral, Qwen 2).

**Weaknesses:**

The major shortcoming of this paper is that the writing style is extremely terse and therefore hard to digest. For example, there are many, many semi-colons stitching together sentence fragments with dense technical buzzwords. For example, at line 122: "Representation-space debiasing (e.g., INLP; adversarial objectives) removes attribute subspaces or reduces recoverability in hidden states (Ravfogel et al., 2020; 2022; Elazar & Goldberg, 2018); such global projections are prompt-agnostic and may erase legitimate, context-linked semantics or miss non-linear effects (Liang et al., 2023) and typically still need weight updates." There were many such sentences like this that took me several reads to parse.

I strongly suspect that this is because almost all of the paper was written or rephrased by a frontier reasoning LLM (I say frontier because ultimately the text does make sense if one spends enough time parsing it, and reasoning because the writing's terseness seems like it could have emerged from reasoning RL with a length penalty).

There are several features of the paper that seem to support this claim, including the terseness and ubiquitous use of semicolons, issues in spacing, the layout of Figure 1, and a few sentences with strange content such as "Main text focuses on representative hardware/settings to conserve space while keeping conclusions intact" (line 421). That being said, I was not able to find definitive proof of this claim.

**Questions:**

Though the paper appears to be technically sound with an interesting method and thorough experiments, the writing is very hard to parse due to its terseness. I would appreciate if the authors could commit to a significant rewrite to make the text more digestible, and would disclose any use of LLMs in drafting the text (as far as I can see, this was not disclosed in the submitted draft).

---

> ### Author Response · Authors · 2025-11-18
> **New paper with enhanced presentation uploaded**
>
> Thank you very much for your detailed review and for recognizing both the **soundness** and the **contribution** of our work. We took your feedback on presentation very seriously and invested substantial effort in revising the paper to make it clearer, less terse, and more visually accessible, while preserving the core technical content.
>
> Below we summarize the concrete changes made in response to your comments.
>
> ---
>
> ### 1. Substantial rewrite of exposition and style
>
> You noted that the writing was “extremely terse,” with many dense, semicolon-heavy sentences that were hard to parse and felt LLM-like. We have:
>
> - **Rewritten the introduction and related work** (Sec. 1–2) to:
>   - Break long, multi-clause sentences into shorter, more readable ones.
>   - Replace chains of semicolons with simpler sentence structure.
>   - Clearly separate motivation, problem setting, and contributions, each in its own concise paragraph.
>
> - **Reorganized Sec. 3 (Method)** around the three-stage workflow in Fig. 1:
>   - Stage I: masking / counterfactual construction
>   - Stage II: dual forward passes and logit fusion
>   - Stage III: dual-branch conformal filtering and sampling
>
> - **Clarified theoretical statements and proofs** (Sec. 3.4–3.5 and App. C):
>   - Explicitly state assumptions and what each theorem guarantees (token-level vs sequence-level).
>   - Provide brief sketches in the main text, with full details in the appendix, to avoid overwhelming the reader.
>
> Overall, we aimed for a more comprehensive academic style: shorter sentences, explicit transitions (“First,… Second,…”) and fewer compressed clauses.
>
> ---
>
> ### 2. Improved figures and visual presentation
>
> You also flagged layout issues (e.g., Fig. 1) as evidence of LLM-heavy drafting. We have:
>
> - **Redesigned Figure 1** into a clean, three-stage pipeline diagram that mirrors the textual structure:
>   - Each stage is visually separated and labeled (“Stage I/II/III”).
>   - The figure caption explains the workflow step-by-step in simple language.
>
>
> - **Adjusted citation and reference appearance** to improve readability:
>   - Citations and references are now colored (e.g., blue) and visually consistent, making them easier to track while reading.
>   - Hyperlinks are clearly visible but not distracting.
>
> We hope this addresses your concern that the paper’s visual design felt “off” or LLM-generated.
>
> ---
>
> ### 3. Statement on the use of large language models
>
> You asked us to disclose any use of LLMs in drafting the text. We now include a dedicated section:
>
> - **“Statement on the use of large language models”** in **Appendix F (last page)**:
>   - We explicitly state that LLMs were used only to assist with **writing polish and local phrasing**.
>   - All core ideas, method design, theoretical results, experiments, analyses, and conclusions were devised and implemented by the authors.
>   - Final revisions, particularly in response to your feedback, were carefully reviewed and edited by the authors to ensure clarity, correctness, and appropriate style.
>
> We hope this directly addresses your concern and makes the authorship process transparent.
>
> ---
>
> Given your positive assessment of **soundness (“good”)** and **contribution (“good”)**, we kindly ask you to reconsider whether the **presentation score** could now better reflect the revised version:
>
> - The exposition has been significantly reworked to be less terse and easier to follow.
> - The method and guarantees are now clearly aligned with Fig. 1 and the three-stage structure.
> - The visual presentation (figures, citations, appendix organization) has been substantially improved.
>
> ---
> Finally, if there are specific sections or passages that you still find hard to read or would like to see simplified further, we would very much appreciate your pointers. We are happy to refine the presentation further wherever possible within the page and formatting constraints.
>
> Thank you again for your thoughtful review, for recognizing the soundness and usefulness of COFT, and for pushing us to make the paper clearer and more readable.

---

> ### Author Response · Authors · 2025-11-22
> **Reviewer accusation**
>
> We would like to respond clearly to the ethics concern raised about our use of large language models (LLMs) and to correct several factual misunderstandings.
>
> ---
>
> ### 1. LLM use was disclosed in the original submission
>
> In the submission form, we explicitly checked:
>
> > **Large Language Models: Yes, to aid or polish writing.**
>
> This disclosure was present from the *initial* submission, not added later. Our use of LLMs was strictly limited to phrasing and polishing, and fully within the scope that ICLR allows for editorial assistance. At no point were core ideas, methods, proofs, or experiments generated by an LLM.
>
> ---
>
> ### 2. Scope of LLM assistance: editorial only, not scientific
>
> All scientific contributions (the COFT method, theoretical results, experimental design, implementation, and analysis) are entirely authored and implemented by us. LLMs were used only to smooth wording and reduce editing time.
>
> In the revised paper we have gone further and added an explicit in-paper disclosure (*“Statement on the Use of Large Language Models”*, Appendix F) which:
>
> - reiterates that LLMs were used only to aid writing, and
> - states that we take full responsibility for all technical content and claims.
>
> ---
>
> ### 3. We addressed the reviewer’s presentation concerns in good faith
>
> In their original review, the reviewer wrote that:
>
> > “the paper appears to be technically sound with an interesting method and thorough experiments,”
>
> and focused their criticism on terseness and readability.
>
> In response, we:
>
> - substantially rewrote the text to reduce terseness, remove overuse of semicolons, and clarify the exposition,
> - redesigned Figure 1 and improved the overall visual presentation, and
> - added the explicit LLM-use statement for maximum transparency.
>
> The revised manuscript is significantly clearer and more readable, while preserving the technical content that the reviewer themselves judged to be sound and interesting.
>
> ---
>
> ### 4. On the ethics accusation, score change, and fairness
>
> After these changes, the reviewer:
>
> - **lowered** their score from 4 (“marginally below acceptance”) to 2,
> - while still agreeing that the paper is technically sound and has an interesting method and thorough experiments, and
> - is now recommending desk rejection on the basis of alleged “unethical” LLM use and “lack of disclosure”.
>
> We respectfully but firmly disagree with this characterization:
>
> - LLM use was explicitly disclosed at submission time (“Yes, to aid or polish writing”).
> - The concern is about the *style* of the initial draft, not about plagiarism, dual submission, fabrication, or any misuse of LLMs to generate scientific content.
> - Penalizing the paper with a drastic score drop and a desk-reject recommendation despite acknowledged technical soundness and contribution, and despite prior disclosure and subsequent clarifications, seems disproportionate and, in our view, unfair.
>
> ---
>
> We remain fully committed to research integrity and transparency. We hope this clarifies the nature and extent of our LLM use and the efforts we took to respond constructively to the reviewer’s concerns.

---

> > ### Comment · Reviewer_DYEN · 2025-11-25
> > **Clarification from AC needed---if the authors disclosed LLM use, why was this not visible to reviewers?**
> >
> > Thanks to the authors for their prompt response.
> >
> > I was surprised to see the authors mention that they *did* disclose their use of LLMs in their original submission. As far as I can tell, this was *not made visible to reviewers*.
> >
> > If the AC confirms that indeed the authors *did* indicate their use of LLMs in the original submission, I would happily retract my claim that the authors have committed an ethics violation and adjust my score to a 4. Then the question is, why does the ICLR OpenReview platform not notify reviewers when the authors have indicated LLM use?
> >
> > On the other hand, if the AC finds that the authors *did not* indicate their use of LLMs in the original submission, then I will maintain my score as a 2 and would recommend desk rejection in line with the stated ICLR policy (https://blog.iclr.cc/2025/11/19/iclr-2026-response-to-llm-generated-papers-and-reviews/).
> >
> > Assuming good intentions from the authors, I apologize for any misunderstanding. I hope that indeed the issue here was that the authors' LLM disclosure was not made visible to reviewers, for some reason.

---

> > > ### Author Response · Authors · 2025-11-25
> > > **Thank you for your follow-up**
> > >
> > > Thank you for your follow-up and for your willingness to clarify this.
> > >
> > > We understand that this may not have been surfaced clearly to reviewers by the platform, and we would very much appreciate it if the AC could confirm this for you as soon as possible. If it helps the process, we are happy to privately share anonymized screenshots of the original submission page showing this field.
> > >
> > > To be completely clear: our use of LLMs was limited to *editorial polishing only*; all ideas, methods, proofs, and experiments are by the authors. In the revised version we have also added an explicit statement in **Appendix F** to avoid any ambiguity going forward.
> > >
> > > We also took your earlier feedback on presentation very seriously: we substantially rewrote several sections for readability, simplified and cleaned up Figure 1, and reduced overly terse or “compressed” phrasing while keeping the technical content intact.
> > >
> > > Since in your initial review you described the paper as technically sound with an interesting method and thorough experiments, and mainly raised concerns about readability and presentation, we would be very grateful if you could let us know if there is anything further we can improve (either in clarity, structure, or specific parts of the exposition) that, in your view, would move the paper into the acceptance range.
> > >
> > > Thank you again for your time and for engaging carefully with both the paper and the discussion.

---

### Meta-Review · Area_Chair_ch33 · 2026-01-11

**Summary:**

The paper presents a test-time method to debias the output of an LLM, which essentially relies on combining a masked & unmasked generation using conformal prediction. Two of the reviewers were mildly positive (and one of them, after the discussion period, became positive) and two of the reviewers were mildly negative.

The reviewers brought a number of concerns, most prominently, there was a consensus that the low quality of the writing/exposition impairs understanding (independently of whether a LLM was used or not to polish the writing) and it is not up to the standards expected at ICLR. The authors tried to address the concern regarding presentation/writing by rewriting the paper during the rebuttal period, however, I think the concern was strong enough to deserve a resubmission.

**Reviewer Concerns:**

The concern regarding the quality of the writing/presentation is still outstanding.

**Reviewer Scores:**

The authors disclosed how the reviewers changed their score.

---

### Decision · Program_Chairs · 2026-01-26

Reject